# Taming "data-hungry" reinforcement learning? Stability in continuous state-action spaces

**Yaqi Duan**
Department of Technology, Operations, and Statistics
Stern School of Business, New York University
New York, NY 10012
yaqi.duan@stern.nyu.edu

**Martin J. Wainwright**
Laboratory for Information and Decision Systems, Statistics and Data Science Center
Department of Electrical Engineering and Computer Science, and Department of Mathematics
Massachusetts Institute of Technology Cambridge, MA 02139
wainwrigwork@gmail.com

## Abstract

We introduce a novel framework for analyzing reinforcement learning (RL) in continuous state-action spaces, and use it to prove fast rates of convergence in both off-line and on-line settings. Our analysis highlights two key stability properties, relating to how changes in value functions and/or policies affect the Bellman operator and occupation measures. We argue that these properties are satisfied in many continuous state-action Markov decision processes. Our analysis also offers fresh perspectives on the roles of pessimism and optimism in off-line and on-line RL.

## 1 Introduction

Many domains of science and engineering involve making a sequence of decisions over time, with previous decisions influencing the future in uncertain ways [1, 13, 22, 31, 32]. For instance, clinicians managing diabetes [36] or engineers optimizing plasma control in tokamak systems [5] must develop policies that adapt based on evolving conditions and lead to desirable outcomes over a longer period. Markov decision processes (MDPs) and reinforcement learning (RL) provide frameworks and methods for estimating effective policies for such sequential problems. While RL excels in data-rich scenarios such as competitive gaming (e.g., AlphaGo and its extensions [30]), its application in data-scarce areas like healthcare [36] and finance [27] remains challenging due to lack of history, or underlying non-stationarity. With limited data, characterizing and improving the *sample complexity* of RL methods becomes critical.

Considerable research effort has been devoted to studying RL sample complexity in many settings. Existing studies for either the generative or the off-line settings (e.g., [21, 37, 35]) give procedures that, when applied to a dataset of size $n$, yield a value gap that decays at the rate $1/\sqrt{n}$. In the on-line setting, there are various procedures that yield cumulative regret that grows at the rate $\sqrt{T}$ (e.g., [18, 20, 19, 6]). In contrast, the main result of this paper is to formalize conditions, suitable for RL in continuous domains, under which *much faster rates can be obtained using the same dataset*, achieving a value gap decay of $1/n$ and reducing regret growth to $\log T$.

As revealed by our analysis, these accelerated rates depend on certain *stability properties*, ones that—as we argue—are naturally satisfied in many control problems with continuous state-action

38th Conference on Neural Information Processing Systems (NeurIPS 2024).

spaces. Roughly speaking, these conditions ensure that the evolution of the dynamic system depends in a "smooth" way on the influence of decision policy. Such notions of stability should be expected in various controlled systems with continuous state-action spaces. In robotics, for example, a minor torque or motion perturbation that occurs during a single step should not cause a notable deviation from the intended trajectory. Similarly, in clinical treatment, slight deviations in medication dosage should not significantly compromise effectiveness or safety.

## 1.1 A simple illustrative example: Mountain Car

The "Mountain Car" problem, a benchmark continuous control task, illustrates the acceleration phenomenon and underlying stability. In this task, as shown in Figure 1(a), a car must reach the top of a hill by adjusting its acceleration within the interval $[-1, 1]$. We employed fitted $Q$-iteration (FQI) with carefully selected linear basis functions to derive near-optimal policies with off-line data. This learning procedure exhibits a value sub-optimality decay at a rate of $1/n$, a significant improvement over the classical rate of $1/\sqrt{n}$, as detailed in Figure 1(b). (See Appendix D for further explanation. The experiment ran for 3 days on two laptops, each equipped with an Apple M2 Pro CPU and 16 GB RAM.) In this example, slight perturbations in the driving policy lead to only modest changes in future trajectories, which shows the stability. Our theoretical analysis confirms that fast rates are achievable in this and similar continuous control tasks when such stability properties are present.

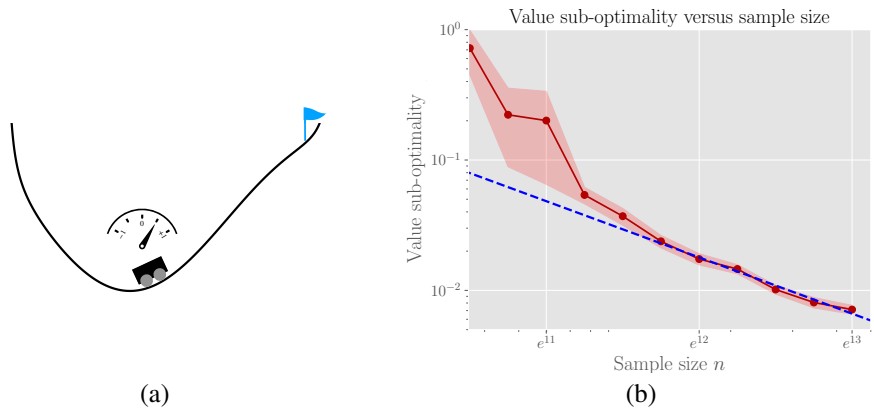

(a)  (b)

Figure 1: Illustration of the "fast rate" phenomenon using FQI on the Mountain Car problem. Each **red point** in the plot represents the average value sub-optimality $J(\pi^\star) - J(\widehat{\pi}_n)$ from $T = 80$ Monte Carlo trials, with the shaded area showing twice the standard errors. The **blue dashed line** is a least-squares fit to the last 6 data points, yielding a $95\%$ confidence interval of $(-1.084, -0.905)$ for the slope, significantly faster than the typical $-0.5$ "slow rate".

## 1.2 Contributions of this paper

With this high-level perspective in mind, let us summarize the key contributions of this paper.

**Fast rate of convergence:** We develop a framework for analyzing RL in continuous state-action spaces, and use it to prove a general result (Theorem 1) under which fast rates can be obtained. The key insight is that stability conditions lead to upper bounds on the value sub-optimality that are proportional to the *squared* norm of Bellman residuals. In the off-line setting, this quadratic scaling improves convergence from a rate of $n^{-\frac{1}{2}}$ to $n^{-1}$, while in on-line learning, it enhances the regret bound from $\sqrt{T}$ to $\log T$.

**Reconsidering pessimism and optimism principles:** Our framework provides a novel perspective on the roles of pessimism [21, 4] and optimism [18, 20, 19, 6, 12] in off-line and on-line RL. Our theory reveals that there are settings in which *neither pessimism nor optimism* are required for effective policy optimization—in particular, they are not required as long as one has a sufficiently accurate pilot estimate policy. Moreover, our analysis shows that some procedures based on certainty

equivalence can achieve fast-rate convergence, showing that the benefits gained from incorporating additional pessimism or optimism measures may be limited in this context.

## 1.3  Related work

In this section, we discuss related work having to do with fast rates in optimization and statistics.

**Fast rates in stochastic optimization and risk minimization:**   For many statistical estimators (e.g., likelihood methods, empirical risk minimization), it is well-understood that the local geometry around the optimum determines whether fast rates can be obtained. For instance, when the loss function exhibits some form of strong convexity (such as exp-concave loss) or strict saddle properties, it can lead to significant reductions in additive regret from $\mathcal{O}(\sqrt{T})$ to just $\mathcal{O}(\log T)$ in stochastic approximation (e.g., [15]), or a decrease in the error rate from $n^{-\frac{1}{2}}$ to $n^{-1}$ in empirical risk minimization [23, 14]. These fast rate phenomena rely on a form of stability, one which relates the similarity of functions to the closeness of their optima. Our work develops a new framework for analyzing value-based RL methods, focusing on identifying specific stability conditions and inherent curvature properties that promote fast rate convergence in RL, similar to the role of stability analysis in statistical learning.

**Fast rates in reinforcement learning:**   In the RL literature, there are various lines of work related to fast rates, but the underlying mechanisms are typically different from those considered here. For problems with discrete state-action spaces, there is a line of recent work [17, 16, 33, 25] that performs gap/marginal-dependent analyses of RL algorithms. However, such separation assumptions are not helpful for continuous action spaces. Other work for discrete state-action spaces [28] has shown convergence rates in off-line RL are influenced by data quality, with a nearly-expert dataset enabling faster rate. In contrast, our analysis reveals that for off-line RL in continuous domains, fast convergence can occur whether or not the dataset has good coverage properties.

An important sub-class of continuous state-action problems are those with linear dynamics and quadratic reward functions (LQR for short). For such problems, it has been shown [24, 29] that value sub-optimality can be connected with the squared error in system identification. Our general theory can also be used to derive guarantees for LQR problems, as we explore in more detail in a follow-up paper [8]. Stability also arises in the analysis of (deterministic) policy optimization and Newton-type algorithms [26, 3], where it is possible to show superlinear convergence in a local neighborhood. This accelerated rate stems from the smoothness of the on-policy transition operator $\mathcal{P}^{\pi_f}$ with respect to changes in the value function $f$; for instance, see condition (10) in Puterman and Brumelle [26]. Our framework exploits related notions of smoothness, but is tailored to the stochastic setting of reinforcement learning, in which understanding the effect of function approximation and finite sample sizes is essential.

## 2  Fast rates for value-based reinforcement learning

Let us now set up and state the main result of this paper. We begin in Section 2.1 with background on Markov decision processes (MDPs) and value-based methods, before turning to the statement of our main result in Section 2.2. In Section 2.3, we provide intuition for why stability leads to faster rates, and discuss consequences for both the off-line and on-line settings of RL.

### 2.1  Markov decision processes and value-based methods

**Basic set-up:**   We consider an episodic Markov decision process (MDP) defined by a quadruple $\big(\mathcal{S}, \mathcal{A}, \mathcal{P} = \{\mathcal{P}_h\}_{h=1}^{H-1}, \{r_h\}_{h=1}^{H}\big)$. We assume that the rewards $r_h : \mathcal{S} \times \mathcal{A} \to \mathbb{R}$ are known; however, this condition can be relaxed. A policy $\pi_h$ at time $h$ is a mapping from any state $s$ to a distribution $\pi_h(\cdot \mid s)$ over the action space $\mathcal{A}$. If the support of $\pi_h(\cdot \mid s)$ is a singleton, we also let $\pi_h(s) \in \mathcal{A}$ denote the single action to be chosen at state $s$. Given an initial distribution $\xi_1$ over the states at time $h = 1$, the *expected reward* obtained by choosing actions according to a policy sequence $\boldsymbol{\pi} = (\pi_1, \ldots, \pi_H)$ is given by $J(\boldsymbol{\pi}) \equiv J(\boldsymbol{\pi}; \xi_1) := \mathbb{E}_{\xi_1, \boldsymbol{\pi}}\big[\sum_{h=1}^{H} r_h(S_h, A_h)\big]$, where $S_1 \sim \xi_1$, $S_{h+1} \sim \mathcal{P}_h(\cdot \mid S_h, A_h)$ and $A_h \sim \pi_h(\cdot \mid S_h)$ for $h = 1, 2, \ldots, H$. Our goal is to estimate an *optimal policy* $\boldsymbol{\pi}^\star \in \arg\max_{\boldsymbol{\pi}} J(\boldsymbol{\pi})$.

**Value functions and Bellman operators:** Starting from a given state-action pair $(s, a)$ at stage $h$, the expected return over subsequent stages defines the *state-action value function* $Q_h^{\boldsymbol{\pi}}(s, a) := \mathbb{E}_{\boldsymbol{\pi}}\big[\sum_{h'=h}^{H} r(S_{h'}, A_{h'}) \mid S_h = s, A_h = a\big]$. The sequence of functions $\boldsymbol{Q}^{\boldsymbol{\pi}} = (Q_1^{\boldsymbol{\pi}}, \ldots, Q_H^{\boldsymbol{\pi}})$ known as the *Q-functions* associated with $\boldsymbol{\pi}$.

The $Q$-functions $\boldsymbol{Q}^{\boldsymbol{\pi}}$ have an important connection with the *Bellman evaluation operator* for $\boldsymbol{\pi}$. For any policy $\boldsymbol{\pi}$ and stage $h$, we introduce a linear transition operator $(\mathcal{P}_h^{\boldsymbol{\pi}} f)(s, a) := \int_{\mathcal{S} \times \mathcal{A}} f(s', a') \, \mathcal{P}_h(ds' \mid s, a) \, \pi_{h+1}(da' \mid s')$ for any function $f \in \mathbb{R}^{\mathcal{S} \times \mathcal{A}}$. With this notation, the *Bellman evaluation operator* at stage $h$ takes the form

$$(\mathcal{T}_h^{\boldsymbol{\pi}} f)(s, a) := r_h(s, a) + (\mathcal{P}_h^{\boldsymbol{\pi}} f)(s, a). \tag{1}$$

From classical dynamic programming, the $Q$-functions $\boldsymbol{Q}^{\boldsymbol{\pi}}$ must satisfy the Bellman relations $Q_h^{\boldsymbol{\pi}}(s, a) = (\mathcal{T}_h^{\boldsymbol{\pi}} Q_{h+1}^{\boldsymbol{\pi}})(s, a)$ for $h = 1, \ldots, H - 1$.

*Bellman principle for optimal policies:* Under mild regularity conditions, there is at least one policy $\boldsymbol{\pi}^\star$ such that, for any other policy $\boldsymbol{\pi}$, we have $Q_h^{\boldsymbol{\pi}^\star}(s, a) \geq Q_h^{\boldsymbol{\pi}}(s, a)$, for any $h \in [H]$, and uniformly over all state-action pairs $(s, a)$. Any optimal policy $\boldsymbol{\pi}^\star$ must be greedy with respect to the optimal $Q$-function $\boldsymbol{Q}^\star$. By classical dynamic programming, the optimal $Q$-function $\boldsymbol{Q}^\star$ is obtained by setting $Q_H^\star = r_H$, and then recursively computing $Q_h^\star = \mathcal{T}_h^\star Q_{h+1}^\star$ for $h = H - 1, \ldots, 2, 1$, with the *Bellman optimality operator* defined as

$$(\mathcal{T}_h^\star f)(s, a) := r_h(s, a) + \mathbb{E}_h\big[\max_{a' \in \mathcal{A}} f(S', a') \mid s, a\big] \qquad \text{for } S' \sim \mathcal{P}_h(\cdot \mid s, a). \tag{2}$$

**Value-based RL methods** The main result of this paper applies to a broad class of methods for reinforcement learning. They are known as *value-based*, due to their reliance on the following two step approach for approximating an optimal policy $\boldsymbol{\pi}^\star$: (1) Construct an estimate $\widehat{\boldsymbol{Q}} = (\widehat{f}_1, \ldots, \widehat{f}_H)$ of the optimal value function $\boldsymbol{Q}^\star = (Q_1^\star, \ldots, Q_H^\star)$. (2) Use $\widehat{\boldsymbol{Q}}$ to compute the greedy-optimal policy $\widehat{\pi}_h(s) \in \arg\max_a \widehat{f}_h(s, a)$ for $h = 1, 2, \ldots, H$. It should be noted that there is considerable freedom in the design of a value-based method, since different methods can be used to approximate value functions in Step 1. Rather than applying to a single method, our main result applies to a very broad class of these methods.

Underlying any value-based method is a class $\mathscr{F}$ of functions $(s, a) \mapsto f(s, a)$ used to approximate the state-action value functions.[1] We assume that the function class $\mathscr{F}$ is rich enough—relative to the Bellman evaluation operators—to ensure that for any greedy policy $\boldsymbol{\pi}$ induced by some $\boldsymbol{f} = (f_1, \ldots, f_H) \in \mathscr{F}^H$, we have the inclusion $\mathcal{T}_h^{\boldsymbol{\pi}} \mathscr{F} \subseteq \mathscr{F}$ for $h = 1, \ldots, H-1$. We see that this condition depends on the structure of the transition distributions $\mathcal{P}_h(\cdot \mid s, a)$. In many practical examples, the reward function itself has some number of derivatives, and these transition distributions perform some type of smoothing, so that we expect that the output of the Bellman update, given a suitably differentiable function, will remain suitably differentiable.

## 2.2 Stable problems have fast rates

We now turn the central question in understanding the behavior of any value-based method:

> *How to translate "closeness" of the $Q$-function estimate $\widehat{\boldsymbol{Q}}$*
> *to a bound on the value gap $J(\boldsymbol{\pi}^\star) - J(\widehat{\boldsymbol{\pi}})$?*

At a high level, existing theory provides guarantees of the following type: if the $Q$-function estimates are $\varepsilon$-accurate for some $\varepsilon \in (0, 1)$, then the value gap is bounded by a quantity proportional to $\varepsilon$. In contrast, our main result shows that when the MDP is stable in a suitable sense, the value gap can be upper bounded by a quantity proportional to $\varepsilon^2$. This *quadratic as opposed to linear scaling* encapsulates the "fast rate" phenomenon of this paper.

Our analysis isolates two key stability properties required for faster rates; both are Lipschitz conditions with respect to a certain norm. Here we define them with respect to the $L^2$-norm induced by the

---

[1] In general, different function classes may be selected at each stage $h = 1, 2, \ldots, H$; here, so as to reduce notational clutter, we assume that the same function class $\mathscr{F}$ is used for each stage.

state-action occupation measure induced by the optimal policy—namely

$$\|f\|_h := \sqrt{\mathbb{E}_{\boldsymbol{\pi}^\star}\left[f^2(S_h, A_h)\right]} \qquad \text{for any } f \in \partial \mathscr{F}^2, \tag{3}$$

and over a neighborhood $\mathcal{N}$ of the optimal $Q$-value function $\boldsymbol{Q}^\star$.

**Bellman stability:** The first condition measures the stability of the Bellman optimality operator (2): in particular, we require that there is a scalar $\kappa_h^\star$ such that

$$\left\|\mathcal{T}_h^\star f_{h+1} - \mathcal{T}_h^\star Q_{h+1}^\star\right\|_h \leq \kappa_h^\star \left\|f_{h+1} - Q_{h+1}^\star\right\|_{h+1} \tag{Stb($\mathcal{T}$)}$$

for any $\boldsymbol{f} \in \mathcal{N}$. Moreover, for any pair $(h, h')$ of indices such that $1 \leq h < h' \leq H - 1$, we define

$$\boldsymbol{\kappa}_{h,h'}(\mathcal{T}^\star) := \kappa_h^\star \, \kappa_{h+1}^\star \ldots \kappa_{h'-1}^\star \,.$$

Condition **(Stb($\mathcal{T}$))** is directly linked to the stability of estimating the $Q$-function $\boldsymbol{Q}^\star$. In typical estimation procedures, such as approximate dynamic programming, the estimation is carried out iteratively in a backward manner, so that it is important to control the propagation of estimation errors across the iterations. Condition **(Stb($\mathcal{T}$))** captures this property, since it implies that

$$\left\|\mathcal{T}_h^\star \mathcal{T}_{h+1}^\star \ldots \mathcal{T}_{h'-1}^\star f_{h'} - \mathcal{T}_h^\star \mathcal{T}_{h+1}^\star \ldots \mathcal{T}_{h'-1}^\star Q_{h'}^\star\right\|_h \leq \boldsymbol{\kappa}_{h,h'}(\mathcal{T}^\star) \cdot \left\|f_{h'} - Q_{h'}^\star\right\|_{h'},$$

which shows how the estimation error $\left(f_{h'} - Q_{h'}^\star\right)$ at step $h'$ can be controlled in terms of estimation error at an earlier time step $h \leq h'$.

**Occupation measure stability:** Our second condition is more subtle, and is key in our argument. Let us begin with some intuition. Consider two sequences of policies

$$\left(\pi_1^\star, \ldots, \pi_{h-1}^\star, \pi_h^\star, \pi_{h+1}^\star, \ldots, \pi_{h'}^\star\right) \qquad \text{and} \qquad \left(\pi_1^\star, \ldots, \pi_{h-1}^\star, \pi_h, \pi_{h+1}^\star, \ldots, \pi_{h'}^\star\right)$$

that only differ at the $h$-th step, where $\pi_h^\star$ has been replaced by $\pi_h$. These two policy sequences induce Markov chains whose distributions differ from stage $h$ onwards, and our second condition controls this difference in terms of the difference $\|f_h - Q_h^\star\|_h$ between the two $Q$-functions $f_h$ and $Q_h^\star$ that induce $\pi_h$ and $\pi_h^\star$, respectively.

We adopt $\mathcal{P}_h^\star$ as a convenient shorthand for the transition operator $\mathcal{P}_h^{\boldsymbol{\pi}^\star}$, and define the multi-step transition operator $\mathcal{P}_{h,h'}^\star := \mathcal{P}_h^\star \mathcal{P}_{h+1}^\star \cdots \mathcal{P}_{h'-1}^\star$. Using this notation, for any $h' \geq h + 1$, we require that there is a scalar $\boldsymbol{\kappa}_{h,h'}(\boldsymbol{\pi}^\star)$ such that

$$\sup_{\substack{g \in \partial \mathscr{F} \\ \|g\|_{h'} > 0}} \frac{\left|\mathbb{E}_{\boldsymbol{\pi}^\star}\left[\left(\mathcal{P}_{h,h'}^\star g\right)(S_h, \pi_h^\star(S_h)) - \left(\mathcal{P}_{h,h'}^\star g\right)(S_h, \pi_h(S_h))\right]\right|}{\|g\|_{h'}} \leq \boldsymbol{\kappa}_{h,h'}(\boldsymbol{\pi}^\star) \frac{\|f_h - Q_h^\star\|_h}{\|Q_h^\star\|_h} \tag{Stb($\xi$)}$$

for any $\boldsymbol{f} \in \mathcal{N}$. The renormalization in this definition serves to enforce a natural scale invariance.

With these notions of stability in hand, we are now equipped to state our main result. Taking as input a value function estimate $\widehat{\boldsymbol{Q}}$, it relates the induced value gap to the *Bellman residuals* $\mathcal{T}_h^\star \widehat{f}_{h+1} - \widehat{f}_h$. Note that these residuals are a way of quantifying proximity to the optimal value function $\boldsymbol{Q}^\star$, which has Bellman residual zero by definition. We assume that $\widehat{\boldsymbol{Q}}$ has Bellman residuals bounded as

$$\left\|\mathcal{T}_h^\star \widehat{f}_{h+1} - \widehat{f}_h\right\|_h \leq \varepsilon_h \qquad \text{for } h = 1, 2, \ldots, H - 1 \tag{4a}$$

for some sequence $\boldsymbol{\varepsilon} = (\varepsilon_1, \ldots, \varepsilon_{H-1}, \varepsilon_H = 0)$ that satisfies the constraint

$$\varepsilon_h \geq \frac{1}{H - h} \sum_{h'=h+1}^{H} \varepsilon_{h'} \qquad \text{for } h = 1, 2, \ldots, H - 1. \tag{4b}$$

This last condition means that the Bellman residual $\varepsilon_h$ is larger than or equal to the average of the bounds established after step $h + 1$. It is natural because estimating at step $h$ is at least as challenging as a stage $h' > h$; indeed, any such state $h'$ occurs earlier in the dynamic programming backward iteration process. As a special case, the bound (4b) holds when $\varepsilon_h = \varepsilon$ for all stages.

With this set-up, we have the following guarantee in terms of the stability coefficients $\boldsymbol{\kappa}_{h,h'}(\boldsymbol{\pi}^\star)$ and $\boldsymbol{\kappa}_{h,h'}(\mathcal{T}^\star)$ from conditions **(Stb($\xi$))** and **(Stb($\mathcal{T}$))**.

---

[2]We let $\partial \mathscr{F}$ be the set of all difference functions of the form $g = f - \widetilde{f}$ for some $f, \widetilde{f} \in \mathscr{F}$.

**Theorem 1.** *There is a neighborhood of $Q^\star$ such that for any value function estimate $\widehat{f}$ with $\varepsilon$-bounded Bellman residuals* (4a)*, the induced greedy policy $\widehat{\pi}$ has value gap bounded as*

$$J(\boldsymbol{\pi}^\star) - J(\widehat{\boldsymbol{\pi}}) \;\leq\; 2 \sum_{h=1}^{H-1} \frac{1}{\|Q_h^\star\|_h} \left\{ \sum_{h'=h}^{H-1} \boldsymbol{\kappa}_{h,h'}(\boldsymbol{\pi}^\star)\, \varepsilon_{h'} \right\} \left\{ \sum_{h'=h}^{H-1} \boldsymbol{\kappa}_{h,h'}(\mathcal{T}^\star)\, \varepsilon_{h'} \right\}. \tag{5}$$

See Appendix A for the proof.

Treating dependence on the stability coefficients as constant, the main take-away is that value sub-optimality is bounded above by a quantity proportional to the *squared* norm of the Bellman residuals. Concretely, if the Bellman residuals are uniformly upper bounded by some $\varepsilon$, then equation (5) leads to an upper bound of the form

$$J(\boldsymbol{\pi}^\star) - J(\widehat{\boldsymbol{\pi}}) \leq c\, H^3\, \varepsilon^2,$$

where $c$ is a universal constant. Due to the quadratic scaling in the Bellman residual error $\varepsilon$, this bound is substantially tighter than the linear in $\varepsilon$ rates afforded by a conventional analysis.

### 2.3 Intuition for fast rates: Smoothness and cancelling terms in the telescope bound

Why does "fast rate" phenomenon formalized in Theorem 1 arise? The fast rates proved in this paper are established by a novel argument, starting from a known telescope bound, which we begin by stating. Given a $Q$-function estimate $\widehat{\boldsymbol{f}} = (\widehat{f}_1, \ldots, \widehat{f}_H)$, let $\widehat{\pi}$ denote the induced greedy policy. Then the value gap of $\widehat{\pi}$ with respect to an arbitrary comparator policy $\pi$ is bounded as

$$J(\boldsymbol{\pi}) - J(\widehat{\boldsymbol{\pi}}) \;\leq\; \sum_{h=1}^{H-1} \left(\mathbb{E}_{\boldsymbol{\pi}} - \mathbb{E}_{\widehat{\boldsymbol{\pi}}}\right) \left[\left(\mathcal{T}_h^\star \widehat{f}_{h+1} - \widehat{f}_h\right)(S_h, A_h)\right]. \tag{6}$$

This result follows by a "telescope" relation induced by the structure of the Bellman updates.[3] For completeness, we provide a proof of the telescope bound in Appendix E.2.

A key feature of inequality (6) is the difference of two expectations $\mathbb{E}_{\boldsymbol{\pi}} - \mathbb{E}_{\widehat{\boldsymbol{\pi}}}$, corresponding to the occupation measures under $\pi$ versus $\widehat{\pi}$. In standard uses of this inequality, an initial argument is used to guarantee that one of these expectations is negative, and so can be dropped [20, 21].

In contrast, the proof of our Theorem 1 exploits a more refined approach, one that handles the difference of expectations directly. Doing so can be beneficial—and lead to "fast rates"— because various terms in this difference can cancel each other out. Specifically, under the smoothness conditions that underlie Theorem 1, when applying the telescope inequality (6) with comparator $\pi = \pi^\star$, we show that the discrepancy between the occupation measures associated with $\pi^\star$ and $\widehat{\pi}$ is of the *same order* as the Bellman residual associated with $\widehat{f}$. Note that the Bellman residuals of $\widehat{f}$ already appear on the right-hand side of inequality (6), so that this fortuitous cancellation can be exploited—along with a number of auxiliary results laid out in the proof—so as to upper bound the value gap by a quantity proportional to the squared Bellman residual $\varepsilon^2$.

It is worthwhile making an explicit comparison of our cancellation approach with the more standard uses of the telescope relation, which typically consider only one portion of the Bellman residuals (e.g., [18, 20, 21, 19, 6, 12, 35]). We do so in the following two subsections.

#### 2.3.1 Pessimism for off-line RL

In the off-line instantiation of RL, the goal is to learn a "good" policy based on a pre-collected dataset $\mathcal{D}$. Note that no further interaction with the environment is permitted, hence the notion of the learning being off-line. More precisely, an *off-line dataset* $\mathcal{D}$ of size $n$ consists of quadruples

$$\mathcal{D} = \left\{ \left(s_{h,i}, a_{h,i}, s'_{h,i}, r_{h,i}\right) \right\}_{i=1}^n,$$

where $s_{h,i}$ and $a_{h,i}$ represent the $i$-th state and action at the $h$-th step in the MDP; $s'_{h,i}$ is the successive state; and $r_{h,i} = r_h(s_{h,i}, a_{h,i})$ denotes the scalar reward. Note that while the successive

---

[3]Results of this type are known; for example, analogous results can be found in past work (e.g., Theorem 2 of the paper [34]; or Lemma 3.2 in the paper [7]).

states are defined by transition dynamics, and the rewards by the reward function, there are no restrictions on how the state-action pairs $(s_{h,\,i}, a_{h,\,i})$ are collected. That is, they need not have been generated by any fixed policy, but may have collected from some ensemble of behavioral policies, or even adaptively by human experts. The goal of off-line reinforcement learning is to use the $n$-sample dataset $\mathcal{D}$ so as to estimate a policy $\widehat{\pi} \equiv \widehat{\pi}_n$ that (approximately) maximizes the expected return $J(\widehat{\pi}_n)$. We expect that—at least for a sensible method for estimating $\widehat{\pi}_n$—the value gap $J(\pi^\star) - J(\widehat{\pi}_n)$ should decay to zero as $n$ increases to infinity, and we are interested in understanding this rate of decay.

The use of pessimism is standard in off-line RL algorithms. Its purpose is to mitigate risks associated with "poor coverage" of the off-line dataset. For instance, the naive approach of simply maximizing $Q$-function estimates based on an off-line dataset can behave poorly when certain portions of the state-action space are not well covered by the given dataset. The pessimism principle suggests to form a *conservative estimate* of the value function—say with

$$\widehat{f}_h(s, a) \leq \mathcal{T}_h^\star \, \widehat{f}_{h+1}(s, a) \tag{7a}$$

with high probability over state-action pairs $(s, a)$. Thus, the estimated value $\widehat{f}_h(s, a)$ is an underestimate of the Bellman update, a form of conservatism that protects against unrealistically high estimates due to poor coverage. Doing so in the appropriate way ensures that

$$-\mathbb{E}_{\widehat{\pi}}\big[\big(\mathcal{T}_h^\star \, \widehat{f}_{h+1} - \widehat{f}_h\big)(S_h, A_h)\big] \leq 0. \tag{7b}$$

Applying this upper bound to the inequality (6) yields the sub-optimality bound

$$J(\pi) - J(\widehat{\pi}) \;\leq\; \sum_{h=1}^{H-1} \mathbb{E}_{\pi}\big[\big(\mathcal{T}_h^\star \, \widehat{f}_{h+1} - \widehat{f}_h\big)(S_h, A_h)\big].$$

Upper bounds derived in this manner only contain one portion of the Bellman residual. When the value functions are approximated in a parametric way (e.g., tabular problems, linear function approximation), this line of analysis leads to value sub-optimality decaying at a "slow" $1/\sqrt{n}$ rate in terms of the sample size $n$ (e.g., [21]). In contrast, an application of Theorem 1 can lead to value gaps bounded by $1/n$.

### 2.3.2 Optimism in on-line RL

In the setting of on-line RL, a learning agent interacts with the environment in a sequential manner, receiving feedback in the form of rewards based on its actions. At the beginning, the learner possesses no prior knowledge of the system's dynamics. In the $t$-th episode, the agent learns an optimal policy $\widehat{\pi}^{(t)}$ using existing observations, implements the policy and collects data $\big\{\big(s_h^{(t)}, a_h^{(t)}, r_h^{(t)}\big)\big\}_{h=1}^{H}$ from the new episode. In each round, the system starts at an initial state $s_1^{(t)}$ independently drawn from a fixed distribution $\xi_1$.

In this on-line setting, it is common to measure the performance of an algorithm by comparing it, over the $T$ rounds of learning, with an oracle that knows and implements an optimal policy. At each round $t$, we incur the *instantaneous regret* $J(\pi^\star) - J(\widehat{\pi}^{(t)})$, where $\pi^\star$ is any optimal policy. Over $T$ rounds, we measure performance in terms of the *cumulative regret*

$$\text{Regret}\big(\{\widehat{\pi}^{(t)}\}_{t=1}^T\big) := \max_{\text{policy } \pi} \; \sum_{t=1}^{T} \Big\{ J(\pi) - J(\widehat{\pi}^{(t)}) \Big\} \;=\; \sum_{t=1}^{T} \underbrace{\Big\{ J(\pi^\star) - J(\widehat{\pi}^{(t)}) \Big\}}_{\text{Regret at round } t}. \tag{8}$$

In a realistic problem, the cumulative regret of any procedure grows with $T$, and our goal is to obtain algorithms whose regret grows as slowly as possible.

In contrast to off-line RL, the on-line setting allows for exploring state-action pairs that have been rarely encountered; doing so makes sense since they might be associated with high rewards. Principled exploration of this type can be effected via the *optimism principle*: one constructs function estimates such that

$$\widehat{f}_h(s, a) \geq \mathcal{T}_h^\star \, \widehat{f}_{h+1}(s, a) \tag{9a}$$

with high probability over state-action pairs.[4] Note that $\widehat{f}_h(s, a)$ is optimistic in the sense that it is an over-estimate of the Bellman update $\mathcal{T}_h^\star \widehat{f}_{h+1}(s, a)$. In this way, we can ensure that

$$\mathbb{E}_{\boldsymbol{\pi}} \big[ \big( \mathcal{T}_h^\star \widehat{f}_{h+1} - \widehat{f}_h \big)(S_h, A_h) \big] \le 0. \tag{9b}$$

Combining this inequality with the telescope bound (6) allows one to upper bound the regret as

$$\mathrm{Regret}\big(\{\widehat{\boldsymbol{\pi}}^{(t)}\}_{t=1}^T\big) \;=\; \sum_{t=1}^T \big\{ J(\boldsymbol{\pi}^\star) - J(\widehat{\boldsymbol{\pi}}^{(t)}) \big\} \;\le\; \sum_{t=1}^T \sum_{h=1}^{H-1} \mathbb{E}_{\widehat{\boldsymbol{\pi}}^{(t)}} \big[ \big( \widehat{f}_h - \mathcal{T}_h^\star \widehat{f}_{h+1} \big)(S_h, A_h) \big].$$

which only includes a single portion of the Bellman residual. In the case of tabular or linear representations of the $Q$-functions, it results in a regret rate of $\sqrt{T}$ (e.g., see the papers [18, 20]). In contrast, an appropriate use of Theorem 1 leads to regret growing only as $\log(T)$, which corresponds to a much better guarantee.

In summary, then, the fast rates obtained in this paper are based on a different approach than the standard pessimism or optimism principles. Since we deal directly with the difference of expectations in the bound (6), there is no need to nullify either of them through the use of these principles. However, it should be noted that we are assuming smoothness conditions that allow us to control this difference. As we discuss in the sequel, such smoothness conditions rule out certain "hard instances" used in past work on lower bounds (e.g. [18, 20, 21, 37]).

## 3 Consequences for linear function approximation

In this section, we explore some consequences of our general theory when applied to value-based methods using (finite-dimensional) linear function approximation.

Let $\boldsymbol{\phi} : \mathcal{S} \times \mathcal{A} \to \mathbb{R}^d$ be a given feature map on the state-action space, and consider linear expansions of the form $f_{\boldsymbol{w}}(s, a) = \langle \boldsymbol{\phi}(s, a), \, \boldsymbol{w} \rangle \equiv \sum_{j=1}^d w_j \boldsymbol{\phi}_j(s, a)$ where $\boldsymbol{w} \in \mathbb{R}^d$ is a weight vector. We adopt the conventional assumption that $\|\boldsymbol{\phi}(s, a)\|_2 \le 1$ and $r_h(s, a) \in [0, 1]$ for all state-action pairs. Defining the linear function class $\mathscr{F} := \big\{ f_{\boldsymbol{w}} \mid \boldsymbol{w} \in \mathbb{R}^d \big\}$, we note that the Minkowski difference class $\partial \mathscr{F}$ is equal to $\mathscr{F}$.

In our analysis of linear approximation, we make use of the norm $\|f\|_h := \sqrt{\mathbb{E}_{\boldsymbol{\pi}^\star}[f^2(S_h, A_h)]}$, corresponding to $L^2$-norm under the occupation measure induced by the optimal policy $\boldsymbol{\pi}^\star$.

### 3.1 Consequences for off-line RL

We now turn to some implications of Theorem 1 for off-line reinforcement learning. Let us recall the off-line setting: for each $h = 1, \dots, H - 1$, we are given a dataset $\mathcal{D}_h = \big\{ (s_{h,i}, a_{h,i}, s'_{h,i}, r_{h,i}) \big\}_{i=1}^n$ of quadruples, from which we can compute estimates $\widehat{\boldsymbol{f}} = (\widehat{f}_h)_{h=1}^H$ with certain Bellman residuals $\{\varepsilon_h\}_{h=1}^{H-1}$, which then appear in the bound (5). The remaining factors on the right-hand side of inequality (5) do not depend on the dataset itself (but rather on structural properties of the MDP). Consequently, in terms of statistical understanding, the main challenge is to establish high-probability bounds on the Bellman residuals $\{\varepsilon_h\}_{h=1}^{H-1}$ for a particular estimator.

#### 3.1.1 Fitted $Q$-iteration (FQI)

As an illustration, let us analyze the use of *fitted $Q$-iteration* (FQI) for computing estimates of the $Q$-function. At a given stage $h = 1, \dots, H - 1$, we can use the associated data $\mathcal{D}_h$ to define a regularized objective function

$$\mathcal{L}_h(f, g) := \frac{1}{|\mathcal{D}_h|} \Bigg[ \sum_{(s_{h,i}, a_{h,i}, s'_{h,i}, r_{h,i}) \in \mathcal{D}_h} \big\{ f(s_{h,i}, a_{h,i}) - \big( r_{h,i} + \max_{a \in \mathcal{A}} g(s'_{h,i}, a) \big) \big\}^2 \Bigg] + \Lambda_h^2(f).$$

Here $g$ represents the target function from stage $h + 1$, and it defines the targeted responses $y_{h,i}(g) := r_{h,i} + \max_{a \in \mathcal{A}} g(s'_{h,i}, a)$. For a given target $g$, we obtain a $Q$-function estimate for

---

[4]Please refer to, for example, Lemma B.3 in the paper [20] for further details.

stage $h$ by minimizing the functional $f \mapsto \mathcal{L}_h(f, g)$. Given that our objective is defined with a quadratic cost, doing so can be understood as a regression method for estimating the conditional expectation that underlies the Bellman update—viz. $\mathcal{T}_h^\star g(s, a) = \mathbb{E}[y_{h, i}(g) \mid (s_{h, i}, a_{h, i}) = (s, a)]$. Here $\Lambda_h^2(f) = \lambda_h \|\boldsymbol{w}\|_2^2$ for $f = \langle \boldsymbol{\phi}(\cdot), \boldsymbol{w} \rangle$ is a regularizer, with $\lambda_h \geq 0$ being the regularization weight. Given this set-up, we can generate a $Q$-function estimate $\widehat{\boldsymbol{f}} = (\widehat{f}_1, \ldots, \widehat{f}_H)$ by first initializing $\widehat{f}_H = r_H$, and then recursively computing $\widehat{f}_h = \arg\min_{f \in \mathscr{F}} \mathcal{L}_h(f, \widehat{f}_{h+1})$, for $h = H - 1, H - 2, \ldots, 2, 1$.

### 3.1.2 Fast rates for FQI-based estimates

In the analysis here, we assume that the dataset consists of i.i.d. tuples (but this can be relaxed as needed). We now state a corollary of Theorem 1, applicable to value function estimates based on FQI with ridge regression.

**Corollary 1** (Fast rates for ridge-based FQI). *For FQI based on ridge regression, with a sufficiently large sample size $n$ and with suitable choices of the regularization parameters $\{\lambda_h\}_{h=1}^{H-1}$, the bound* (5) *from Theorem 1 holds with*

$$\varepsilon_h = c \sqrt{\{d(H - h)/n\} \log(dH/\delta)} \tag{10}$$

*with probability at least $1 - \delta$.*

We omit the proof of Corollary 1, as it follows from standard ridge regression analysis.

**Fast rates and comparisons to past work:** So as to be able to compare with results from past work, let us consider some consequences of the bound (10) under the following assumptions: (i) $\boldsymbol{\kappa}_{h, h'}(\mathcal{T}^\star) = \mathcal{O}(1)$; (ii) $\boldsymbol{\kappa}_{h, h'}(\boldsymbol{\pi}^\star) = \mathcal{O}(\sqrt{d})$; (iii) $\|Q_h^\star\|_h \asymp H - h + 1$. Then it can be shown that the bound from Corollary 1 takes the form

$$J(\boldsymbol{\pi}^\star) - J(\widehat{\boldsymbol{\pi}}) \leq c \, d^{3/2} \, H^3 \, n^{-1} \, \log(dH/\delta), \tag{11a}$$

and is valid for a sample size $n \geq c d^2 H^3$. Alternatively stated, Corollary 1 guarantees that for FQI using ridge regression with $d$-dimensional function approximation, the number of samples $n(\epsilon)$ required to obtain $\epsilon$-optimal policy is at most

$$n_{\text{fast}}(\epsilon) \asymp d^{\frac{3}{2}} H^3 / \epsilon + d^2 H^3, \tag{11b}$$

where we use $\asymp$ to denote a scaling that ignores constants and logarithmic factors.

Let us compare this guarantee to related work by Zanette et al. [37], who analyzed the use of pessimistic actor-critic methods for linear function classes. When translated into the notation of our paper, their analysis established[5] a sample complexity of the order $n_{\text{Zan}}(\epsilon) \asymp d^2 H^3 / \epsilon^2$. Consequently, we see that once the target error $\epsilon$ is relatively small—$\epsilon \in (0, 1)$—then stable MDPs can exhibit a much smaller $(1/\epsilon)$ sample complexity.

It should be noted that past work (e.g., [21, 37]) has established $(1/\epsilon^2)$-lower bounds on the sample complexity of estimating $\epsilon$ policies in the off-line setting. However, these lower bounds *do not* contradict our fast rate guarantee (11b), because the "hard instances" used in these lower bound proofs violate the stability condition (**Stb**($\xi$)). In particular, even infinitessimally small perturbations in policy lead to occupation measures that are significantly different.

**When is pessimism necessary?** An interesting aspect of the guarantee from Corollary 1 is that it provides guarantees for off-policy RL (and with fast rates) using a method that does *not* incorporate any form of pessimism. This is a sharp contrast with many other methods for off-policy RL, such as pessimistic forms of $Q$-learning and actor-critic methods (e.g., [21, 37]).

To be clear, as noted following the bound (11a), the guarantee from Corollary 1 requires the sample size to be lower bounded as $n \geq c d^2 H^3$. In contrast, pessimistic schemes only require a sample size sufficiently large to ensure validity of the Bellman residual upper bounds that underlie Corollary 1— meaning that $n \gtrsim d$ up to logarithmic factors. Thus, the pessimism principle can be useful for problems with smaller sample sizes.

---

[5]See Appendix C.2 for the details of this calculation.

## 3.2 Consequences for on-line RL

In this section, we explore some consequences of Theorem 1 for on-line reinforcement learning. We begin by describing a two-stage procedure[6] that allows us to convert the risk bounds for FQI from off-line RL into regret in on-line RL:

**Phase 1** (*Exploration*) In the initial $T_0$f episodes, the focus is purely on exploration, resulting in an estimate of $Q$-function denoted as $\widehat{\boldsymbol{f}}^{(T_0)}$.

**Phase 2** (*Fine-tuning*) For $k = 0, 1, \ldots, K-1$ with $K := \lceil \log_2(T/T_0) \rceil$, repeat:

- In the $t$-th episode, for each $t = T_0\, 2^k + 1, \ldots, T_0\, 2^{k+1}$, execute the greedy policy induced by function $\widehat{\boldsymbol{f}}^{(T_0\, 2^k)}$.
- Update the $Q$-function estimate $\widehat{\boldsymbol{f}}^{(T_0\, 2^{k+1})}$ using FQI based on observations collected from episodes $T_0\, 2^k + 1, T_0\, 2^k + 2, \ldots, T_0\, 2^{k+1}$.

We assume the burn-in time $T_0$ is large enough so as to ensure the pilot $Q$-function estimate $\widehat{\boldsymbol{f}}^{(T_0)}$ obtained in Phase 1 falls within a certain "absorbing" region $\mathcal{N}(\boldsymbol{\rho})$ around $\boldsymbol{Q}^\star$. Under these conditions, we have the following bound on the regret.

**Corollary 2.** *For FQI based on ridge regression with rewards in* $[0, 1]$*, with a sufficiently large burn-in time* $T_0$ *and with suitable choices of the regularization parameters* $\{\lambda_h\}_{h=1}^{H-1}$*, the two-phase scheme achieves regret bounded as*

$$Regret(T) \leq c\left\{ T_0 \cdot H + d\sqrt{d}\, H^4 \log T \cdot \log(dHK/\delta) \right\}$$

*with probability at least* $1 - \delta$.

See Appendix C.1 for the proof.

**Sharper bound on regret:** The leading term (as $T$ grows) in the regret bound grows as $\log T$, which is much smaller than the typical $\sqrt{T}$-rate found in past work [18, 20]. The $\sqrt{T}$ rate has been shown to be unimprovable in general, but the worst-case instances[18, 20] that lead to $\sqrt{T}$-regret violate the stability conditions used in our analysis.

**When is optimism needed?** The use of optimism—by adding bonuses to the current value function estimates so as to encourage exploration—underlies many schemes in on-line RL. An interesting take-away from Corollary 2 is that under the stability conditions highlighted by our theory, it is possible to achieve excellent regret bounds without the use of optimism. In our two-phase scheme, the only exploration occurs in Phase 1. All other data is simply collected using the greedy policy induced by the current $Q$-function estimate. A well-designed exploration scheme—one that might incorporate the optimism principle—is necessary only during the burn-in Phase 1.

## 4 Discussion

This paper introduces a novel approach for the analysis of value-based RL methods for continuous state-action spaces. Our analysis highlights two key stability properties of MDPs under which much sharper bounds on value sub-optimality can be guaranteed. Our analysis offers fresh perspectives on the commonly used pessimism and optimism principles, in off-line and on-line settings respectively.

Our study leaves open various questions for future work. First, our main result (Theorem 1) has consequences for linear quadratic control, to be described in an upcoming paper [8]. It provides insight into the role of covariate shift in linear quadratic control, as well as efficient exploration in the on-line setting. Second, our current statistical analysis focused on i.i.d. data with linear function approximation. It is interesting to consider the extensions to dependent data and non-parametric function approximation (e.g. kernels, boosting, and neural networks). Third, while this paper has provided upper bounds, it remains to address the complementary question of lower bounds for policy optimization over the classes of stable MDPs isolated here. Last, to better align our framework with real-world scenarios, we intend to go beyond the idealized completeness condition used in this paper, and treat the role of model mis-specification.

---

[6]To be clear, the purpose of this scheme is primarily conceptual, rather than practical in nature.

**Acknowledgements**

This work was partially supported by NSF grant CCF-1955450, ONR grant N00014-21-1-2842, and NSF DMS-2311072 to MJW.

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

# A  Proofs of Theorem 1

This section is devoted to the proof of Theorem 1, which consists of three main steps. These steps rely on two auxiliary lemmas whose proofs are fairly technical, so that they are deferred to in Appendices B.1 and B.2.

**High-level outline:**   Let us outline the three steps of the proof. In Step 1, we use a one-step expansion of the difference in the occupation measures to reformulate the standard telescope inequality (6). Doing so results in a relation with structure similar to that of the left-hand side of inequality (**Stb**($\xi$)). In Step 2, we develop a constraint on the function estimation error $d_h(\widehat{f}_h, Q_h^\star)$ that ensures the occupation measure produced by policy $\widehat{\pi}$ remains stable and does not deviate too much from the occupation measure associated with the optimal policy $\pi^\star$. In Step 3, we use Bellman stability (**Stb**($\mathcal{T}$)) to connect the $Q$-function error $\widehat{f}_h - Q_h^\star$ with Bellman residuals. With this high-level view in place, we now work through the three steps.

## A.1  Step 1: Reformulation of the telescope inequality.

Recall the standard telescope inequality (6). Our proof makes use of an alternative form, which involves the functions

$$\Delta_h(\boldsymbol{\pi};\, s, a\,) \;=\; \sum_{h'=h}^{H-1} \mathcal{P}_{h,h'}^{\boldsymbol{\pi}}\big(\mathcal{T}_{h'}^\star \widehat{f}_{h'+1} - \widehat{f}_{h'}\big)(s, a). \tag{12}$$

**Lemma 1.** *Given a Q-function estimate $\widehat{\boldsymbol{f}} = \big(\widehat{f}_1, \ldots, \widehat{f}_{H-1}, \widehat{f}_H = r_H\big)$ and the associated greedy policy $\widehat{\pi}$, we have the bound*

$$J(\boldsymbol{\pi}) - J(\widehat{\boldsymbol{\pi}}) \;\leq\; \sum_{h=1}^{H-1} \mathbb{E}_{\widehat{\boldsymbol{\pi}}}\big[\Delta_h(\boldsymbol{\pi};\, s_h, \pi_h(s_h)) - \Delta_h(\boldsymbol{\pi};\, s_h, \widehat{\pi}_h(s_h))\big] \tag{13}$$

*valid for any policy $\boldsymbol{\pi}$.*

See Appendix B.1 for the proof.

We apply the bound (13) with $\boldsymbol{\pi} = \boldsymbol{\pi}^\star$. Following some algebra, we find that

$$J(\boldsymbol{\pi}^\star) - J(\widehat{\boldsymbol{\pi}}) \;\leq\; \sum_{h=1}^{H-1}\sum_{h'=h}^{H-1} \widehat{\beta}(h,\, h') \cdot \varepsilon_{h'}\,,$$

where $\varepsilon_{h'}$ is an upper bound on the Bellman residual $\big\|\mathcal{T}_{h'}^\star \widehat{f}_{h'+1} - \widehat{f}_{h'}\big\|_{h'}$ as given in equation (4a). The term $\widehat{\beta}(h,\, h')$ is given by

$$\widehat{\beta}(h,\, h') := \sup_{f \in \partial\mathscr{F}:\, \|f\|_{h'} > 0} \left\{ \frac{1}{\|f\|_{h'}} \Big| \mathbb{E}_{\widehat{\boldsymbol{\pi}}}\Big[\big(\mathcal{P}_{h,h'}^\star f\big)(s_h, \pi_h^\star(s_h)) - \big(\mathcal{P}_{h,h'}^\star f\big)(s_h, \widehat{\pi}_h(s_h))\Big]\Big| \right\}. \tag{14a}$$

We note that the left-hand side of inequality (**Stb**($\xi$)) has a similar form to the term $\widehat{\beta}(h,\, h')$, differing only in that the expectation is taken over the occupation measure of running the optimal policy $\boldsymbol{\pi}^\star$, rather than the estimated policy $\widehat{\boldsymbol{\pi}}$.

## A.2  Step 2: Constraint to ensure stability

Our next step is to establish an upper bound on the coefficient $\widehat{\beta}(h, h')$ defined by the estimated policy $\widehat{\pi}$ in terms of the analogous quantity defined by the optimal policy $\pi^\star$—namely, the coefficient

$$\beta(h, h') := \sup_{f \in \partial\mathscr{F}:\, \|f\|_{h'} > 0} \left\{ \frac{1}{\|f\|_{h'}} \Big| \mathbb{E}_{\boldsymbol{\pi}^\star}\Big[\big(\mathcal{P}_{h,h'}^\star f\big)(s_h, \pi_h^\star(s_h)) - \big(\mathcal{P}_{h,h'}^\star f\big)(s_h, \widehat{\pi}_h(s_h))\Big]\Big| \right\}. \tag{14b}$$

In order to do so, we demonstrate that a sufficiently small function estimation error $d_h(\widehat{f}_h, Q_h^\star)$ ensures the inequality

$$\sum_{h=1}^{H-1} \sum_{h'=h}^{H-1} \widehat{\beta}(h, h') \cdot \varepsilon_{h'} \;\leq\; 2 \sum_{h=1}^{H-1} \sum_{h'=h}^{H-1} \beta(h, h') \cdot \varepsilon_{h'} \, . \tag{15}$$

Once we have established this bound, we can replace the term $\beta(h, h')$ with $\kappa_{h,h'}(\boldsymbol{\pi}^\star) \cdot \big\| \widehat{f}_h - Q_h^\star \big\|_h / \|Q_h^\star\|_h$, using the inequality **(Stb($\xi$))**.

We summarize the result in the following auxiliary lemma:

**Lemma 2.** *Suppose that the function estimation errors satisfy* $d_h(Q_h, Q_h^\star) \leq \frac{1}{2\, b_{\mathscr{F}}} (H - h + 1)^{-1}$ *for* $h = 2, 3, \ldots, H - 1$ *and the sequence* $\boldsymbol{\varepsilon} = (\varepsilon_1, \ldots, \varepsilon_{H-1}, \varepsilon_H = 0)$ *satisfies the regularity condition* (4b)*. Then we have*

$$J(\boldsymbol{\pi}^\star) - J(\widehat{\boldsymbol{\pi}}) \;\leq\; 2 \sum_{h=1}^{H-1} \frac{\big\| \widehat{f}_h - Q_h^\star \big\|_h}{\|Q_h^\star\|_h} \left\{ \sum_{h'=h}^{H-1} \kappa_{h,h'}(\boldsymbol{\pi}^\star) \, \varepsilon_{h'} \right\}. \tag{16}$$

See Appendix B.2 for the proof.

### A.3 Step 3: Connecting $Q$-function error and Bellman residuals

The remaining piece of the proof is to connect the function difference $\widehat{f}_h - Q_h^\star$ with Bellman residuals $\mathcal{T}_h^\star \widehat{f}_{h+1} - \widehat{f}_h$, using the stability condition **(Stb($\mathcal{T}$))** on the Bellman operator $\mathcal{T}^\star$. This is relatively straightforward: indeed, we claim that

$$\big\| \widehat{f}_h - Q_h^\star \big\|_h \;\leq\; \sum_{h'=h}^{H-1} \kappa_{h,h'}(\mathcal{T}^\star) \cdot \big\| \mathcal{T}_{h'}^\star \widehat{f}_{h'+1} - \widehat{f}_{h'} \big\|_{h'} \, . \tag{17}$$

Recall that $Q_h^\star = \mathcal{T}_h^\star Q_{h+1}^\star$ for $h = 1, 2, \ldots, H - 1$. Therefore, we have

$$\widehat{f}_h - Q_h^\star = \big( \mathcal{T}_h^\star \widehat{f}_{h+1} - \mathcal{T}_h^\star Q_{h+1}^\star \big) - \big( \mathcal{T}_h^\star \widehat{f}_{h+1} - \widehat{f}_h \big) \, .$$

By employing the triangle inequality and the Bellman stability given in equation **(Stb($\mathcal{T}$))**, we derive that

$$\begin{aligned}
\big\| \widehat{f}_h - Q_h^\star \big\|_h &\leq \big\| \mathcal{T}_h^\star \widehat{f}_{h+1} - \widehat{f}_h \big\|_h + \big\| \mathcal{T}_h^\star \widehat{f}_{h+1} - \mathcal{T}_h^\star Q_{h+1}^\star \big\|_h \\
&\leq \big\| \mathcal{T}_h^\star \widehat{f}_{h+1} - \widehat{f}_h \big\|_h + \kappa_h^\star \big\| \widehat{f}_{h+1} - Q_{h+1}^\star \big\|_{h+1} \, .
\end{aligned}$$

Applying this inequality recursively yields the claim (17).

With this piece in place, we can complete the proof of Theorem 1. Indeed, we have

$$\begin{aligned}
J(\boldsymbol{\pi}^\star) - J(\widehat{\boldsymbol{\pi}}) &\overset{(a)}{\leq} 2 \sum_{h=1}^{H-1} \frac{\big\| \widehat{f}_h - Q_h^\star \big\|_h}{\|Q_h^\star\|_h} \left\{ \sum_{h'=h}^{H-1} \kappa_{h,h'}(\boldsymbol{\pi}^\star) \, \varepsilon_{h'} \right\} \\
&\overset{(b)}{\leq} 2 \sum_{h=1}^{H-1} \frac{1}{\|Q_h^\star\|_h} \left\{ \sum_{h'=h}^{H-1} \kappa_{h,h'}(\mathcal{T}^\star) \, \varepsilon_{h'} \right\} \left\{ \sum_{h'=h}^{H-1} \kappa_{h,h'}(\boldsymbol{\pi}^\star) \, \varepsilon_{h'} \right\}.
\end{aligned}$$

Here step (a) is a restatement of the bound (16) from Lemma 2, whereas step (b) follows from inequality (17). Thus, we have established the claim given in Theorem 1.

## B  Proof of auxiliary lemmas for Theorem 1

We now turn to proofs of the two auxiliary results used to establish our main theorem, with Lemmas 1 and 2 treated in Appendices B.1 and B.2, respectively.

## B.1 Proof of Lemma 1

For any integrable vector function $\boldsymbol{g} = (g_1, \ldots, g_H) \in \mathbb{R}^{\mathcal{S} \times \mathcal{A} \times H}$, we define

$$D(\boldsymbol{g}) \; = \; \sum_{h=1}^{H} \left( \mathbb{E}_{\boldsymbol{\pi}} - \mathbb{E}_{\widehat{\boldsymbol{\pi}}} \right) \left[ g_h(S_h, A_h) \right]. \tag{18a}$$

We claim that this functional satisfies the recursive relation

$$D(\boldsymbol{g}) = \sum_{h=1}^{H} \mathbb{E}_{\widehat{\boldsymbol{\pi}}} \left[ g_h(S_h, \pi_h(S_h)) - g_h(S_h, \widehat{\pi}_h(S_h)) \right] + D(\boldsymbol{\mathcal{P}}^{\boldsymbol{\pi}} \boldsymbol{g}), \tag{18b}$$

where we have introduced the shorthand $\boldsymbol{\mathcal{P}}^{\boldsymbol{\pi}} \boldsymbol{g} := \left( \mathcal{P}_1^{\boldsymbol{\pi}} g_2, \ldots, \mathcal{P}_{H-1}^{\boldsymbol{\pi}} g_H, 0 \right) \in \mathbb{R}^{\mathcal{S} \times \mathcal{A} \times H}$.

Taking this claim as given for the moment, let us prove the bound (13) from Lemma 1. First, we set $\boldsymbol{g} := (\boldsymbol{\mathcal{P}}^{\boldsymbol{\pi}})^h \, \boldsymbol{g} = \left( \mathcal{P}_{1,1+h}^{\boldsymbol{\pi}} \, g_{1+h}, \ldots, \mathcal{P}_{H-h,H}^{\boldsymbol{\pi}} \, g_H, 0, \ldots, 0 \right)$ in equation (18b) for $h = 0, 1, \ldots, H-1$, which yields

$$D\big( (\boldsymbol{\mathcal{P}}^{\boldsymbol{\pi}})^h \, \boldsymbol{g} \big) = \sum_{\substack{1 \le h' \le j \le H, \\ j - h' = h}} \mathbb{E}_{\widehat{\boldsymbol{\pi}}} \left[ \{ \mathcal{P}_{h',j}^{\boldsymbol{\pi}} \, g_j \}(S_{h'}, \pi_{h'}(S_{h'})) - \{ \mathcal{P}_{h',j}^{\boldsymbol{\pi}} \, g_j \}(S_{h'}, \widehat{\pi}_{h'}(S_{h'})) \right]$$
$$+ D\big( (\boldsymbol{\mathcal{P}}^{\boldsymbol{\pi}})^{h+1} \, \boldsymbol{g} \big).$$

Note that $(\boldsymbol{\mathcal{P}}^{\boldsymbol{\pi}})^H \, \boldsymbol{g} = 0$, which implies $D\big( (\boldsymbol{\mathcal{P}}^{\boldsymbol{\pi}})^H \, \boldsymbol{g} \big) = 0$. We then sum the resulting bounds so as to obtain

$$D(\boldsymbol{g}) = \sum_{1 \le h \le h' \le H} \mathbb{E}_{\widehat{\boldsymbol{\pi}}} \left[ \{ \mathcal{P}_{h,h'}^{\boldsymbol{\pi}} \, g_{h'} \}(S_h, \pi_h(S_h)) - \{ \mathcal{P}_{h,h'}^{\boldsymbol{\pi}} \, g_{h'} \}(S_h, \widehat{\pi}_h(S_h)) \right]. \tag{19}$$

Setting $\boldsymbol{g} = \boldsymbol{\mathcal{T}}^{\star} \widehat{\boldsymbol{f}} - \widehat{\boldsymbol{f}}$, or equivalently $g_h = \mathcal{T}_h^{\star} \, \widehat{f}_{h+1} - \widehat{f}_h$, in equation (19), we find that

$$D\big( \boldsymbol{\mathcal{T}}^{\star} \widehat{\boldsymbol{f}} - \widehat{\boldsymbol{f}} \big) = \sum_{h=1}^{H-1} \mathbb{E}_{\widehat{\boldsymbol{\pi}}} \left[ \Delta_h(\boldsymbol{\pi}; S_h, \pi_h(S_h)) - \Delta_h(\boldsymbol{\pi}; S_h, \widehat{\pi}_h(S_h)) \right],$$

where we have used the fact (12) that $\Delta_h(\boldsymbol{\pi}; \cdot) = \sum_{h'=h}^{H} \mathcal{P}_{h,h'}^{\boldsymbol{\pi}} \big( \mathcal{T}_{h'}^{\star} \, \widehat{f}_{h'+1} - \widehat{f}_{h'} \big)$. Thus, we have established the bound (13) stated in Lemma 1.

It remains to establish the auxiliary claim (18b). Note that the functional $D$ can be decomposed as $D(\boldsymbol{g}) = D_1 + D_2$, where

$$D_1 := \sum_{h=1}^{H} \mathbb{E}_{\widehat{\boldsymbol{\pi}}} \left[ g_h(S_h, \pi_h(S_h)) - g_h(S_h, \widehat{\pi}_h(S_h)) \right] \qquad \text{and}$$

$$D_2 := \sum_{h=1}^{H} \left( \mathbb{E}_{\boldsymbol{\pi}} - \mathbb{E}_{\widehat{\boldsymbol{\pi}}} \right) \left[ g_h(S_h, \pi_h(S_h)) \right].$$

Applying the tower property of conditional expectation, we find that

$$D_2 = \sum_{h=1}^{H-1} \left( \mathbb{E}_{\boldsymbol{\pi}} - \mathbb{E}_{\widehat{\boldsymbol{\pi}}} \right) \left[ \mathbb{E}[g_{h+1}(S_{h+1}, \pi_{h+1}(S_{h+1})) \mid S_h, A_h] \right]$$

$$= \sum_{h=1}^{H-1} \left( \mathbb{E}_{\boldsymbol{\pi}} - \mathbb{E}_{\widehat{\boldsymbol{\pi}}} \right) \left[ (\mathcal{P}_h^{\boldsymbol{\pi}} \, g_{h+1})(S_h, A_h) \right] = D(\boldsymbol{\mathcal{P}}^{\boldsymbol{\pi}} \boldsymbol{g}).$$

Combining the expressions for $D_1$ and $D_2$ above yields the claim (18b).

## B.2 Proof of Lemma 2

The key step in proving Lemma 2 is establishing that inequality (15) holds when the function estimation error $d_h(\widehat{f}_h, Q_h^\star)$ is sufficiently small. In order to do so, we need to establish upper bounds on the term $\widehat{\beta}(h, h')$ by using $\beta(h, h')$. In particular, we will show that for any $1 \leq h \leq h' \leq H - 1$,

$$\widehat{\beta}(h,\, h') \;\leq\; \beta(h,\, h') \;+\; \sum_{j=1}^{h-1} \widehat{\beta}(j,\, h-1) \,\cdot\, b_{\mathscr{F}} \,\cdot\, d_h(\widehat{f}_h,\, Q_h^\star)\,. \tag{20}$$

The inequality (20) is derived based on the definitions of metric $d_h$ and parameter $b_{\mathscr{F}} = 1$. After a close examination of the right-hand side of this inequality, it becomes evident that as long as the function estimation error $d_h(\widehat{f}_h, Q_h^\star)$ is sufficiently small, the terms associated with $d_h(\widehat{f}_h, Q_h^\star)$ are negligible and are dominated by $\beta(h, h')$. Consequently, inequality (15) within the arguments in Appendix A.2 is likely to hold true.

With claim (20) assumed to be valid at this point, we now establish a proper upper bound on the estimation error $d_h(\widehat{f}_h, Q_h^\star)$ under which inequality (15) is satisfied. By taking linear combinations of inequality (20) using weights $\varepsilon = (\varepsilon_1, \ldots, \varepsilon_{H-1}, \varepsilon_H = 0)$, we obtain

$$\sum_{h=1}^{H-1} \sum_{h'=h}^{H-1} \widehat{\beta}(h,\, h') \cdot \varepsilon_{h'} \;\leq\; \sum_{h=1}^{H-1} \sum_{h'=h}^{H-1} \beta(h,\, h') \cdot \varepsilon_{h'}$$
$$+ \sum_{h=2}^{H-1} \sum_{j=1}^{h-1} \widehat{\beta}(j,\, h-1) \,\cdot\, b_{\mathscr{F}} \,\cdot\, d_h(\widehat{f}_h,\, Q_h^\star) \sum_{h'=h}^{H-1} \varepsilon_{h'}\,. \tag{21}$$

When the sequence $\varepsilon = (\varepsilon_1, \ldots, \varepsilon_{H-1}, \varepsilon_H = 0)$ is regular in the sense that inequality (4b) holds, the bound (21) reduces to

$$\sum_{1 \leq h \leq h' \leq H} \widehat{\beta}(h,\, h') \cdot \varepsilon_{h'} \;\leq\; \sum_{1 \leq h \leq h' \leq H} \beta(h,\, h') \cdot \varepsilon_{h'}$$
$$+ \sum_{1 \leq h \leq h' \leq H-2} \widehat{\beta}(h,\, h') \cdot \varepsilon_{h'} \,\cdot\, b_{\mathscr{F}}\, (H - h') \,\cdot\, d_{h'+1}(\widehat{f}_{h'+1},\, Q_{h'+1}^\star)\,.$$

Under the condition $d_h(\widehat{f}_h, Q_h^\star) \leq \frac{1}{2\, b_{\mathscr{F}}} (H - h + 1)^{-1}$ for $2 \leq h \leq H - 1$, the inequality above implies bound (15), which further establishes the bound (16), as stated in Lemma 2.

It remains to prove the relation between $\widehat{\beta}(h,\, h')$ and $\beta(h,\, h')$, as shown in inequality (20).

**Proof of bound** (20): It is evident that inequality (20) holds for $h = 1$, therefore, we focus on its validation for indices $2 \leq h \leq H - 1$. Recall the definitions of functions $\widehat{\beta}(h, h')$ and $\beta(h, h')$, as given by equations (14a) and (14b). We apply the triangle inequality and derive that

$$\left| \widehat{\beta}(h,\, h') - \beta(h,\, h') \right|$$
$$\leq \sup_{f \in \partial\mathscr{F} : \|f\|_{h'} > 0} \left\{ \frac{1}{\|f\|_{h'}} \left| (\mathbb{E}_{\widehat{\pi}} - \mathbb{E}_{\pi^\star}) \left[ (\mathcal{P}_{h,h'}^\star f)(S_h, \pi_h^\star(S_h)) - (\mathcal{P}_{h,h'}^\star f)(S_h, \widehat{\pi}_h(S_h)) \right] \right| \right\}$$
$$= \sup_{f \in \partial\mathscr{F} : \|f\|_{h'} > 0} \left\{ \frac{1}{\|f\|_{h'}} \left| (\mathbb{E}_{\widehat{\pi}} - \mathbb{E}_{\pi^\star}) \left[ \{ (\mathcal{P}_{h-1}^\star - \mathcal{P}_{h-1}^{\widehat{\pi}}) \mathcal{P}_{h,h'}^\star f \}(S_{h-1}, A_{h-1}) \right] \right| \right\}$$
$$=: \Delta\beta(h,\, h')\,.$$

The term $\Delta\beta(h,\, h')$ involves differences from two sources: (i) the difference in transition kernels $\mathcal{P}_{h-1}^\star - \mathcal{P}_{h-1}^{\widehat{\pi}}$ that captures the divergence between policies $\pi_h^\star$ and $\widehat{\pi}_h$; (ii) the discrepancy of occupation measures at the $(h-1)$-th step reflected by the difference in expectations $\mathbb{E}_{\pi^\star} - \mathbb{E}_{\widehat{\pi}}$, which is determined by the policies $(\pi_1^\star, \ldots, \pi_{h-1}^\star)$ and $(\widehat{\pi}_1, \ldots, \widehat{\pi}_{h-1})$ until the $(h-1)$-th step. We treat them separately and write

$$\Delta\beta(h,\, h') \;\leq\; \nu_1(h-1,\, h') \,\cdot\, \nu_2(h-1)\,, \tag{22}$$

where the functionals $\nu_2$ and $\nu_1$ are defined as

$$\nu_1(h-1,\,h') := \sup_{f \in \partial\mathscr{F}:\, \|f\|_{h'} > 0} \left\{ \frac{1}{\|f\|_{h'}} \left\| \left( \mathcal{P}_{h-1}^\star - \mathcal{P}_{h-1}^{\widehat{\pi}} \right) \mathcal{P}_{h,h'}^\star\, f \right\|_{h-1} \right\},$$

$$\nu_2(h-1) := \sup_{f \in \partial\mathscr{F}:\, \|f\|_{h-1} > 0} \left\{ \frac{1}{\|f\|_{h-1}} \left| \left( \mathbb{E}_{\widehat{\pi}} - \mathbb{E}_{\pi^\star} \right) \left[ f(S_{h-1}, A_{h-1}) \right] \right| \right\}.$$

We first consider the term $\nu_1$. According to the definitions of metric $d_h$ and parameter $b_\mathscr{F}$ we find that

$$\left\| \left( \mathcal{P}_{h-1}^\star - \mathcal{P}_{h-1}^{\widehat{\pi}} \right) \mathcal{P}_{h,h'}^\star\, f \right\|_{h-1} \leq d_h\big( \widehat{f}_h, Q_h^\star \big) \cdot \left\| \mathcal{P}_{h,h'}^\star\, f \right\|_h \overset{(*)}{\leq} d_h\big( \widehat{f}_h, Q_h^\star \big) \cdot b_\mathscr{F}\, \|f\|_{h'} \,,$$

which in turn implies

$$\nu_1(h-1,\,h') \;\leq\; b_\mathscr{F} \,\cdot\, d_h\big( \widehat{f}_h, Q_h^\star \big). \tag{23a}$$

The proof of inequality $(*)$ for $b_\mathscr{F} = 1$, as mentioned above, can be found in Appendix E.1.

As for term $\nu_2$, we claim that

$$\nu_2(h-1) \;\leq\; \sum_{j=1}^{h-1} \widehat{\beta}(j,\,h-1) \,. \tag{23b}$$

Combining the bound $\widehat{\beta}(h,\,h') \leq \beta(h,\,h') + \Delta\beta(h,\,h')$ with inequalities (22), (23a) and (23b), we establish the bound (20), as claimed. It remains to prove the claim (23b).

**Proof of inequality** (23b)**:** This proof is analogous to that of Lemma 1. We begin by introducing an analogue of the functional $D(g)$ from equation (18a); in particular, for any index $h \in [H-1]$ and function $g \in \partial\mathscr{F}$, define

$$D_h^\star(g) := \left( \mathbb{E}_{\pi^\star} - \mathbb{E}_{\widehat{\pi}} \right) \left[ g(S_h, A_h) \right].$$

Using the notation of $D_h^\star$, we can rewrite the left-hand side of inequality (23b) as $\nu_2(h-1) = \sup_{f \in \partial\mathscr{F}:\, \|f\|_{h-1} > 0} \left\{ |D_{h-1}^\star(f)| / \|f\|_{h-1} \right\}$. Following the same arguments as in the proof of inequality (18b), we can show that

$$D_h^\star(g) = \mathbb{E}_{\widehat{\pi}} \left[ g(S_h, \pi_h^\star(S_h)) - g(S_h, \widehat{\pi}_h(S_h)) \right] + D_{h-1}^\star(\mathcal{P}_{h-1}^\star g) \qquad \text{for } h = 1, 2, \ldots, H, \tag{24}$$

where we set $D_0^\star \equiv 0$.

We consider function $g := \mathcal{P}_{j,h-1}^\star f$ for $1 \leq j < h \leq H-1$. It follows from equation (24) that

$$D_j^\star\big( \mathcal{P}_{j,h-1}^\star f \big) = \mathbb{E}_{\widehat{\pi}} \left[ \big( \mathcal{P}_{j,h-1}^\star f \big)(S_j, \pi_j^\star(S_j)) - \big( \mathcal{P}_{j,h-1}^\star f \big)(S_j, \widehat{\pi}_j(S_j)) \right] + D_{j-1}^\star\big( \mathcal{P}_{j-1,h-1}^\star f \big),$$

where we have used the relation $\mathcal{P}_{j-1}^\star \mathcal{P}_{j,h-1}^\star = \mathcal{P}_{j-1,h-1}^\star$. Recalling the definition of $\widehat{\beta}(j,\,h-1)$ in equation (14a), applying the triangle inequality yields

$$\left| D_j^\star\big( \mathcal{P}_{j,h-1}^\star f \big) \right| \;\leq\; \widehat{\beta}(j,\,h-1) \cdot \|f\|_{h-1} \,+\, \left| D_{j-1}^\star\big( \mathcal{P}_{j-1,h-1}^\star f \big) \right|.$$

Summing this equation over indices $j = 1, 2, 3, \ldots, h-1$ yields

$$|D_{h-1}^\star(f)| \;\leq\; \sum_{j=1}^{h-1} \widehat{\beta}(j,\,h-1) \cdot \|f\|_{h-1} \,,$$

which establishes inequality (23b).

# C  Proof of corollaries

This section cotains proofs of several corollaries.

## C.1  Proof of Corollary 2

We now turn to proving Corollary 2 regarding ridge-based FQI in on-line settings.

In Phase 1 of pure exploration, the cumulative regret is always bounded from above by $T_0 \cdot H$. During Phase 2 of fine-tuning, we let $\widehat{\boldsymbol{\pi}}^k$ be the policy employed in the rounds $T_0 2^k + 1, T_0 2^k + 2, \ldots,$ $T_0 2^{k+1}$, which is determined by the estimate $\widehat{\boldsymbol{f}}^{(T_0 2^k)}$ calculated at the end of the $(T_0 2^k)$-th round. To estimate the regret, we consider the decomposition

$$\sum_{t=T_0+1}^{T} \left\{ J(\boldsymbol{\pi}^\star) - J(\widehat{\boldsymbol{\pi}}^{(t)}) \right\} \leq \sum_{k=0}^{K-1} \sum_{t=T_0 2^k}^{T_0 2^{k+1}} \left\{ J(\boldsymbol{\pi}^\star) - J(\widehat{\boldsymbol{\pi}}^{(t)}) \right\} = \sum_{k=0}^{K-1} T_0 2^k \left\{ J(\boldsymbol{\pi}^\star) - J(\widehat{\boldsymbol{\pi}}^k) \right\}.$$

We leverage our bound (11a) for off-line RL in Section 3.1.2 to control the value sub-optimality $J(\boldsymbol{\pi}^\star) - J(\widehat{\boldsymbol{\pi}}^k)$. Recall that the policy $\widehat{\boldsymbol{\pi}}^k$ is derived from i.i.d. trajectories collected from the rounds $T_0 2^{k-1} + 1, T_0 2^{k-1} + 2, \ldots, T_0 2^k$. We divide those $T_0 2^{k-1}$ trajectories into $H - 1$ equal shares and use each share to conduct estimation in one iteration of the FQI procedure. This subsampling technique ensures the independence of samples used in different iterations. It is primarily adopted for the sake of convenience (to keep the explanations concise) and is not essential in general. It follows from inequality (11a) that the bound

$$J(\boldsymbol{\pi}^\star) - J(\widehat{\boldsymbol{\pi}}^k) \leq c \, \frac{d\sqrt{d} \, H^4}{T_0 \, 2^k} \, \log(dHK/\delta)$$

holds uniformly for indices $k = 0, 1, \ldots, K-1$ with a probability exceeding $1 - \delta$.

Putting together the pieces, we arrive at

$$\mathrm{Regret}(T) \leq T_0 \cdot H + c \, d\sqrt{d} \, H^4 \, K \, \log(dHK/\delta).$$

We then derive the regret bound in Corollary 2 by noticing that $K = \mathcal{O}(\log T)$.

## C.2  Comparing to known off-line bounds

In this section, we derive the sample complexity $n_{\mathrm{Zan}}(\epsilon) \asymp \frac{d^2 H^3}{\epsilon^2}$ in Section 3.1.2 based on the results of Zanette et al. [37]; it gives the conventional $1/\sqrt{n}$ slow rate to which we compare. Zanette et al. [37] proved upper bounds on a pessimistic actor-critic scheme based on $d$-dimensional linear function approximation. Using our notation, Theorem 1 in their paper [37] can be expressed as

$$J(\boldsymbol{\pi}^\star) - J(\widehat{\boldsymbol{\pi}}) \leq c \left\{ \frac{1}{H} \sum_{h=1}^{H-1} \sqrt{\overline{\boldsymbol{\phi}}_h^\top (\widehat{\boldsymbol{\Sigma}}_{h,\mathcal{D}} + \lambda_h \boldsymbol{I})^{-1} \overline{\boldsymbol{\phi}}_h} \right\} \sqrt{\frac{dH^4}{n}}, \tag{25}$$

where the vector $\overline{\boldsymbol{\phi}}_h$ is given by $\overline{\boldsymbol{\phi}}_h := \mathbb{E}_{\boldsymbol{\pi}^\star} \left[ \boldsymbol{\phi}(S_h, A_h) \right]$, the covariance matrix

$$\widehat{\boldsymbol{\Sigma}}_{h,\mathcal{D}} := \frac{1}{|\mathcal{D}_h|} \sum_{(s_{h,i}, a_{h,i}, s'_{h,i}, r_{h,i}) \in \mathcal{D}_h} \boldsymbol{\phi}(s_{h,i}, a_{h,i}) \boldsymbol{\phi}(s_{h,i}, a_{h,i})^\top.$$

We now consider the explicit dependence of this upper bound on dimension $d$, horizon $H$ and sample size $n$. The divergence term $\overline{\boldsymbol{\phi}}_h^\top (\widehat{\boldsymbol{\Sigma}}_{h,\mathcal{D}} + \lambda_h \boldsymbol{I})^{-1} \overline{\boldsymbol{\phi}}_h$ measures the conditioning of the regularized covariance matrix $(\widehat{\boldsymbol{\Sigma}}_{h,\mathcal{D}} + \lambda_h \boldsymbol{I})$ along a specific direction of $\overline{\boldsymbol{\phi}}_h$. When the feature mapping $\boldsymbol{\phi}$ operates within a $d$-dimensional space, it is reasonable to assume that

$$\overline{\boldsymbol{\phi}}_h^\top (\widehat{\boldsymbol{\Sigma}}_{h,\mathcal{D}} + \lambda_h \boldsymbol{I})^{-1} \overline{\boldsymbol{\phi}}_h \leq c' \, d.$$

The bound (25) then reduces to $J(\boldsymbol{\pi}^\star) - J(\widehat{\boldsymbol{\pi}}) \leq c \, dH^2/\sqrt{n}$. Regarding the dependence on horizon $H$, we conjecture that by incorporating the law of total variance in a more refined manner, it may be possible to further reduce the dependence by a factor of $\sqrt{H}$. Under these conditions, the bound takes the form $J(\boldsymbol{\pi}^\star) - J(\widehat{\boldsymbol{\pi}}) \leq c \, d\sqrt{H^3/n}$.

# D  Details of the mountain car experiment

In this experiment, a car is situated in a valley between two hills. The car's objective is to overcome the gravitational pull and reach the top of the right hill by efficiently controlling its acceleration.

## D.1  Structure of the Markov decision process

The Markov decision process underlying the mountain car problem has a state space $\mathcal{S} \subset \mathbb{R}^2$ and an action space $\mathcal{A} \subset \mathbb{R}$. The state $s = (p, v)$ consists of the current position $p$ and velocity $v$, whereas the scalar action $a = f$ corresponds to the applied input force. The state variables $(p, v)$ and action $f$ are restricted as

$$
\begin{aligned}
p &\in [p_{\min}, p_{\max}] = [-1.2, 0.6], \\
v &\in [v_{\min}, v_{\max}] = [-0.07, 0.07] \quad \text{and} \\
f &\in [f_{\min}, f_{\max}] = [-1, 1].
\end{aligned}
$$

The mountain is described by the function

$$
m(p) = \tfrac{1}{3} \sin(3p) + \frac{0.025}{(p_{\max} - p)(p - p_{\min})},
$$

over the interval $p \in [p_{\min}, p_{\max}]$.

Let $m'$ be the derivative of the mountain shape function $m$, which represents the instantaneous slope, and let $(\sigma_v, \sigma_p) = (0.01, 0.0025)$ be a pair of standard deviations that dictate the amount of randomness in the updates. For an interval $[a, b]$, we define the truncation function

$$
\Psi_{[a,b]}(u) := \begin{cases} u & \text{if } u \in [a, b], \\ b & \text{if } u > b, \\ a & \text{if } u < a. \end{cases}
$$

With this notation, at each discrete time step $h = 0, 1, 2, \ldots$, the position and velocity of the car evolve as

$$
\begin{aligned}
v_{h+1} &= \Psi_{[v_{\min}, v_{\max}]} \big( v_h + 0.0015 \, f_h - 0.0025 \, m'(p_h) + \sigma_v Z_h \big) \\
p_{h+1} &= \Psi_{[p_{\min}, p_{\max}]} \big( p_h + v_{h+1} + \sigma_p Z'_h \big)
\end{aligned}
$$

where $(Z_h, Z'_h)$ are a pair of independent standard normal variables. Note that the system dynamics are non-linear due to both the presence of the derivative $m'$ and the truncation function $\Psi$.

The objective of the car is to reach the peak of the mountain, designated by the position $p_{\text{goal}} = 0.45$. The reward at state-action pair $(s, a)$ is given by

$$
r(s, a) := -\tfrac{1}{10} f^2 + 100 \big[ \max\{0, \, p - p_{\text{goal}}\} \big]^2.
$$

For any policy $\pi$, we define the $\gamma$-discounted value function

$$
J(\pi) := \mathbb{E}_\pi \Big[ \sum_{h=0}^{\infty} \gamma^h \, r(S_h, A_h) \Big],
$$

using $\gamma = 0.97$. The initial state $s_0 = (p_0, v_0)$ is generated with $p_0$ following a uniform distribution over the interval $[-0.6, -0.4]$, and we initialize with velocity $v_0 = 0$.

## D.2  Fitted Q-iteration (FQI) with linear function approximation

Here we describe the use of fitted Q-iteration (FQI) with linear function approximation to estimate the optimal $Q$-function, along with the corresponding greedy policy $\widehat{\pi}$.

**Linear function approximation**  We approximate the the optimal $Q$-function $(s, a) \mapsto Q^\star(s, a)$ using a $d$-dimensional linear function class with $d = 3000$ features. We begin by defining the *base*

*feature maps* $\boldsymbol{\phi}_p : [\, p_{\min}, p_{\max} \,] \to \mathbb{R}^{50}$ for position, and $\boldsymbol{\phi}_v : [\, v_{\min}, v_{\max} \,] \to \mathbb{R}^{15}$ for velocity, with components given by

$$
\begin{cases}
\phi_{p,2j+1}(p) := \cos(jp), & \text{for } j = 0, 1, \ldots, 24, \quad \text{and} \\
\phi_{p,2j}(p) := \sin(jp), & \text{for } j = 1, 2, \ldots, 25\,;
\end{cases}
$$

$$
\begin{cases}
\phi_{v,2j+1}(v) := \cos(jv), & \text{for } j = 0, 1, \ldots, 7, \quad \text{and} \\
\phi_{v,2j}(v) := \sin(jv), & \text{for } j = 1, 2, \ldots, 7.
\end{cases}
$$

To represent the action $a \equiv f$, we define the *base action feature map*

$$
\boldsymbol{\phi}_f(f) := \left(1, f, f^2, f^3\right) \in \mathbb{R}^4.
$$

The overall feature map $\boldsymbol{\phi} : \mathcal{S} \times \mathcal{A} \to \mathbb{R}^{3000}$ is constructed by taking the outer product of the three base feature maps $\boldsymbol{\phi}_p$, $\boldsymbol{\phi}_v$, and $\boldsymbol{\phi}_f$ as follows:

$$
\boldsymbol{\phi}(s, a) := \mathrm{vec}\big\{\boldsymbol{\phi}_p(p) \otimes \boldsymbol{\phi}_v(v) \otimes \boldsymbol{\phi}_f(f)\big\} \in \mathbb{R}^{3000}. \tag{26}
$$

Taking all possible triples of the three base features in the outer product leads to the overall dimension $d = 3000 = 50 \times 15 \times 4$. Given a weight vector $\boldsymbol{w} \in \mathbb{R}^{3000}$, we define the function $f_{\boldsymbol{w}}(s, a) := \langle \boldsymbol{w}, \boldsymbol{\phi}(s, a) \rangle$, and we approximate the optimal $Q$-function using the function class $\mathscr{F} := \big\{f_{\boldsymbol{w}} \mid \boldsymbol{w} \in \mathbb{R}^{3000}\big\}$.

**Fitted Q-iteration (FQI)**   We employed fitted Q-iteration with the linear feature $\boldsymbol{\phi} : \mathcal{S} \times \mathcal{A} \to \mathbb{R}^{3000}$ to estimate an optimal policy $\widehat{\pi}$. The FQI process begins by initializing the weight vector as $\boldsymbol{w}_0 := \boldsymbol{0} \in \mathbb{R}^{3000}$. In each iteration, we first use the dataset $\mathcal{D} = \big\{(s_i, a_i, r_i, s_i')\big\}_{i=1}^n \subset \mathcal{S} \times \mathcal{A} \times \mathbb{R} \times \mathcal{S}$ to construct the pseudo-responses

$$
y_i := r_i + \gamma \max_{a \in \mathcal{A}} \underbrace{\langle \boldsymbol{w}_t, \boldsymbol{\phi}(s_i', a) \rangle}_{f_{\boldsymbol{w}_t}(s_i', a)} \qquad \text{for } i = 1, \ldots, n, \tag{27}
$$

corresponding to a stochastic estimate of the Bellman update applied to our current $Q$-function estimate $f_{\boldsymbol{w}_t}$. The polynomial form of the force feature $\boldsymbol{\phi}_f$ allows for a closed-form solution to the maximum operation required in equation (27). Given these pseudo-responses, we then update the weight vector $\boldsymbol{w}_t \to \boldsymbol{w}_{t+1}$ via the ridge regression

$$
\boldsymbol{w}_{t+1} := \arg\min_{\boldsymbol{w} \in \mathbb{R}^{3000}} \left\{ \frac{1}{n} \sum_{i=1}^n \big\{y_i - \langle \boldsymbol{w}, \boldsymbol{\phi}(s_i, a_i) \rangle\big\}^2 + \lambda_n \|\boldsymbol{w}\|_2^2 \right\}, \tag{28}
$$

where $\lambda_n = \frac{0.01}{n}$ in all experiments reported here.

We terminate the procedure after at most 500 iterations, or when there have been 5 consecutive iterations with insignificant improvements in weights, where insignificant means that $\|\boldsymbol{w}_{t+1} - \boldsymbol{w}_t\|_2 / \sqrt{3000} < 0.005$. Letting $\widehat{\boldsymbol{w}}$ represent the weight vector obtained from this procedure, the resulting policy $\widehat{\pi}$ is given by selecting the greedy action based on the $Q$-function estimate $\widehat{f}(s, a) := \langle \widehat{\boldsymbol{w}}, \boldsymbol{\phi}(s, a) \rangle$.

### D.3   Experimental configurations

Our experiments were based on an off-line dataset consisting of $n$ i.i.d. tuples

$$
\mathcal{D} = \big\{(s_i, a_i, r_i, s_i')\big\}_{i=1}^n \subset \mathcal{S} \times \mathcal{A} \times \mathbb{R} \times \mathcal{S},
$$

where the state-action pairs $\big\{(s_i, a_i) = (p_i, v_i, f_i)\big\}_{i=1}^n$ were generated from a uniform distribution over the cube $[p_{\min}, p_{\max}] \times [v_{\min}, v_{\max}] \times [f_{\min}, f_{\max}]$. We performed independent experiments with the sample size $n$ varying over the range

$$
n \in \big\{\lfloor e^k \rfloor \mid k = 10.5, 10.75, 11, \ldots, 13\big\}
$$
$$
= \{36315, 46630, 59874, 76879, 98715, 126753, 162754, 208981, 268337, 344551, 442413\}.
$$

In each experiment, we generated a dataset $\mathcal{D}$, estimated an optimal policy $\widehat{\pi}$ based on the data, and evaluated the return $J(\widehat{\pi})$. For each sample size, we conducted 80 independent trials.

In order to evaluate the return $J(\widehat{\pi})$, for each initial position $p_0 = -0.5 + 0.2\,j/1000$ with $j = -500, -499, -498, \ldots, 499$, we simulated 30 independent 1000-step trajectories by executing the estimated policy $\widehat{\pi}$. The average return over the $30 \times 1000$ trajectories is used as the estimate of $J(\widehat{\pi})$.

In order to approximate the policy[7] $\pi^\dagger$ that represents "ground truth", we conducted a single experiment with sample size $n = 6.4 \times 10^6$ to obtain $\pi^\dagger$. We simulated 1000 trajectories for each initial position $p_0$ and calculated the average return, which serves as the reference value $J(\pi^\dagger)$. The value sub-optimality is then computed as the difference $J(\pi^\dagger) - J(\widehat{\pi})$.

## E   Verification of auxiliary claims

In this appendix, we collect the verification of various auxiliary claims made in the main text.

### E.1   Properties of occupation measures

In this appendix, we prove a useful inequality

$$\left\| \mathcal{P}_{h,h'}^\star f \right\|_h \le \| f \|_{h'}, \tag{29}$$

which holds for the state-action occupation measures (3). This bound is used in the proof of bound (20) in Appendix B.2.

By definition, we have

$$\left\| \mathcal{P}_h^\star f \right\|_h^2 = \mathbb{E}_{\pi^\star}\left[ (\mathcal{P}_h^\star f)^2 (S_h, A_h) \right] = \mathbb{E}_{\pi^\star}\left[ \mathbb{E}_h\left[ f(S_{h+1}, \pi_{h+1}^\star(S_{h+1})) \mid S_h, A_h \right]^2 \right].$$

According to the property of variance, we can deduce

$$\mathbb{E}_{\pi^\star}\left[ \mathbb{E}_h\left[ f(S_{h+1}, \pi_{h+1}^\star(S_{h+1})) \mid S_h, A_h \right]^2 \right] \le \mathbb{E}_{\pi^\star}\left[ f^2(S_{h+1}, \pi_{h+1}^\star(S_{h+1})) \right] = \| f \|_{h+1}^2.$$

As a consequence, we find that $\left\| \mathcal{P}_h^\star f \right\|_h \le \| f \|_{h+1}$. Applying this inequality recursively leads to the conclusion that for any indices $1 \le h \le h' \le H$, we have

$$\left\| \mathcal{P}_{h,h'}^\star f \right\|_h = \left\| \mathcal{P}_h^\star \mathcal{P}_{h+1,h'}^\star f \right\|_h \le \left\| \mathcal{P}_{h+1,h'}^\star f \right\|_{h+1} \le \left\| \mathcal{P}_{h+2,h'}^\star f \right\|_{h+2} \le \cdots \le \| f \|_{h'}.$$

This establishes the bound (29).

### E.2   Proof of the telescope inequality (6)

For completeness of this paper,[8] let us prove the telescope relation (6) stated in Section 2.3. For any policy $\boldsymbol{\pi} = (\pi_1, \ldots, \pi_H)$ and sequence of functions $\boldsymbol{f} = (f_1, \ldots, f_H)$ with $f_H = r_H$, we have the "telescope" relation

$$V_1^{\boldsymbol{\pi}}(s) = f_1(s, \pi_1(s)) + \sum_{h=1}^{H-1} \mathbb{E}_{\boldsymbol{\pi}}\left[ \left( \mathcal{T}_h^{\boldsymbol{\pi}} f_{h+1} - f_h \right)(S_h, A_h) \mid S_1 = s \right] \quad \text{for any state } s \in \mathcal{S}. \tag{30}$$

Here the value function $V_1^{\boldsymbol{\pi}}$ is given by $V_1^{\boldsymbol{\pi}}(s) := Q_1^{\boldsymbol{\pi}}(s, \pi_1(s))$. Taking $\boldsymbol{f} = \widehat{\boldsymbol{f}}$ in equation (30) yields

$$V_1^{\boldsymbol{\pi}}(s) = \widehat{f}_1(s, \pi_1(s)) + \sum_{h=1}^{H-1} \mathbb{E}_{\boldsymbol{\pi}}\left[ \left( \mathcal{T}_h^{\boldsymbol{\pi}} \widehat{f}_{h+1} - \widehat{f}_h \right)(S_h, A_h) \mid S_1 = s \right]. \tag{31a}$$

Letting $\boldsymbol{\pi} = \widehat{\boldsymbol{\pi}}$ in equation (31a) yields

$$V_1^{\widehat{\boldsymbol{\pi}}}(s) = \widehat{f}_1(s, \widehat{\pi}_1(s)) + \sum_{h=1}^{H-1} \mathbb{E}_{\widehat{\boldsymbol{\pi}}}\left[ \left( \mathcal{T}_h^{\widehat{\boldsymbol{\pi}}} \widehat{f}_{h+1} - \widehat{f}_h \right)(S_h, A_h) \mid S_1 = s \right]. \tag{31b}$$

---

[7] In general, it is not guaranteed that $\pi^\dagger$ is equal to the optimal policy $\pi^\star$, due to approximation error that might arise from using the linear function class defined here.

[8] We are not claiming novelty here; see Theorem 2 of the paper [34]; or Lemma 3.2 in the paper [7] for analogous results.

Since $\widehat{\pi}$ is a greedy policy with respect to function $\widehat{f}$, we have

$$\widehat{f}_1(s, \widehat{\pi}_1(s)) \geq \widehat{f}_1(s, \pi_1(s)), \quad \text{and} \quad \mathcal{T}_h^{\widehat{\pi}} \widehat{f}_{h+1} = \mathcal{T}_h^{\star} \widehat{f}_{h+1} \geq \mathcal{T}_h^{\pi} \widehat{f}_{h+1} \quad \text{for any policy } \pi.$$

Using this fact and subtracting equations (31a) and (31b), we obtain

$$V_1^{\pi}(s) - V_1^{\widehat{\pi}}(s) \leq \sum_{h=1}^{H-1} \left( \mathbb{E}_{\pi} - \mathbb{E}_{\widehat{\pi}} \right) \left[ \left( \mathcal{T}_h^{\star} \widehat{f}_{h+1} - \widehat{f}_h \right)(S_h, A_h) \mid S_1 = s \right].$$

Finally, taking the expectation over the initial distribution $\xi_1$ yields the claimed inequality (6).

