# OpenReview forum: "Taming "data-hungry" reinforcement learning? Stability in continuous state-action spaces"
_NeurIPS.cc/2024/Conference — NeurIPS 2024 poster_

### Official Review · Reviewer_bfUp · 2024-07-01

**Soundness:** 3
**Presentation:** 4
**Contribution:** 4
**Rating:** 7
**Confidence:** 3

**Summary:**

The paper presents a novel approach to deriving convergence using two 'stability' properties or assumptions. Improved bounds are derived which show faster convergence than predicted by traditional bounds.

**Strengths:**

1. The authors present a new framework for analyzing RL convergence properties. The authors propose two new stability notions that will help them derive improved convergence rates: a. propogation of estimation errors across iterations b. change of occupancy measure w.r.t policy

2. The paper uses the above two notions (which are relatively mild) to derive bounds in the standard tabular setting that decay as 1/n for offline and log T in the online setting, which is in contrast to previous bounds of 1/n^2 or \sqrt(T) respectively. This takes a step towards partly explaining the fast convergence of RL algorithms observed in practice, atleast in the environments where the stability properties hold.

3.  The intuition for the proof of fast rates in section 2.3 greatly helps in clarifying the approach towards deriving these rates. Authors extend the analysis to the setting of linear function approximation.

4. The bounds lead to interesting insights on covariate shifts and are justified as purely being a computing the loss under a different distribution of data.

5. Finally, the authors demonstrate an important benefit of their framework - their bounds do not require any kind of pessimism and yet are able to derive fast bounds. It is discussed under what conditions ignoring pessimism or optimism can still give better bounds.

**Weaknesses:**

1. Validating the theory with some MDPs, even toy domains can greatly help in increasing the quality of paper.

**Questions:**

None

**Limitations:**

The authors have discusses limitations of their work in Section 4.

---

> ### Author Rebuttal · Authors · 2024-08-07
>
> Thank you for your positive feedback on our work. We really appreciate it!
>
> - $Weakness$
>
> RE: Thank you for your helpful suggestion. We will add toy examples and include more discussions to improve the clarity of the paper.

---

> ### Comment · Area_Chair_QLuK · 2024-08-11
> **Please respond to the authors**
>
> Hello reviewer bfUp: The authors have responded to your comments. I would expect you to respond in kind.

---

> > ### Comment · Reviewer_bfUp · 2024-08-13
> >
> > I thank the reviewers for their rebuttal. I ll maintain my score of acceptance.

---

### Official Review · Reviewer_PVTM · 2024-07-09

**Soundness:** 3
**Presentation:** 2
**Contribution:** 3
**Rating:** 7
**Confidence:** 2

**Summary:**

In the context of continual control reinforcement learning, this paper presents a new analysis of how a good Q-function estimate (measured in terms of the Bellman residual norms) induces a good greedy policy (measured in terms of the value gap compared to the optimal policy). The key contribution is the formalization of two "stability" criteria under which the authors prove $O(n^{-1})$ convergence of fitted Q-iteration in the offline RL setting and $O(\log T)$ regret in the online setting, considerably faster than previous results which do not make "stability" assumptions.

**Strengths:**

The paper starts with a very illustrative motivating example that empirically demonstrates the fast $O(n^{-1})$ convergence rate of FQI not explained by the known upper bound of $O(n^{-\frac{1}{2}})$. They go on to introduce two "stability" assumptions about MDPs under which they show that this faster convergence rate is to be expected, and even extend this result to the online setting demonstrating that the previous upper bound on the regret of $O(\sqrt T)$ can be improved to $O(\log T)$ under the same stability assumptions. This is a strong result that will be of interest to the community. To the best of my knowledge, the ideas presented in this paper are novel.

**Weaknesses:**

1. This is the most important point. The two assumptions "Bellman stability" and "Occupation measure stability" are very difficult to understand. There is no proper motivation or illustrative example about when to expect these conditions to hold, and when not to expect so. The authors do translate these conditions to the special case of linear function approximation, but also these "specialized" conditions ("Curv1" and "Curv2") are unintuitive. The paper is very full already and I can understand why the authors decided to save space and not include in-depth discussions of these conditions. However, as these assumptions are really at the core of their results and what differentiates their setting from the "slower rate" setting, I believe the paper would be much clearer if, for example, Section 3 were shortened considerably and instead the stability assumptions were explained in more detail.
2. It is not clear to me what the authors mean by "stability". This term has a concrete meaning in control theory, but I don't understand how this meaning is related to their definitions of "Bellman stability" and "Occupation measure stability". This point is closely related to point 1.
3. (minor) The theorem statements should include the full set of assumptions. For example, Theorem 1 should explicitly say that it holds only under the two stability assumptions.
4. (minor) In Figure 1b, the x-axis uses a logarithm base of $e$, while the y-axis uses the base $10$. This is not just unusual, it also makes the discussion about the slope of $-1$ vs. $-\frac{1}{2}$ confusing, as the plot actually shows a slope of $-\frac{1}{\ln 10}$.
5. (minor) The probability notation in line 307 is confusing. Do you mean $\mu_h^\star(s, a) = \mathbb P_{\xi_1, \pi^\star}[S_h = s, A_h = a]$?

**Questions:**

The analysis presented in the paper shows that fitted Q-iteration converges much faster than previous results show if the environment satisfies certain stability criteria. Is it possible to exploit this stability explicitly in a new method, or is the rate achieved by the general FQI method already optimal?

**Limitations:**

The authors adequately address the limitations of their work.

---

> ### Author Rebuttal · Authors · 2024-08-07
>
> Thank you for recognizing the value of our research! Below, we will address each of the points you’ve raised.
>
> - $Weakness 1$
>
> RE: Thank you for your valuable feedback. We will work on shortening Section 3 and adding more discussions of stability conditions and toy examples to make the intuitions more explicit.
>
> Regarding the intuitions behind the two assumptions:
>
> -- Bellman stability: This can be interpreted as the propagation of estimation errors across backward iterations in dynamic programming.
>
> -- Occupation measure stability: This relates to the forward evolution of divergences in occupancy measures after perturbations on policy.
>
> These assumptions are relatively natural in many practical cases: Bellman stability ensures the validity of value function estimation, while occupation measure stability characterizes the sensitivity of the system to policy changes.
>
> Regarding the curvature conditions, we plan to move the geometric illustration from Appendix G to the main body to help readers understand better.
>
> - $Weakness 2$
>
> RE: Thank you for raising this point. We will be more cautious about the use of the term “stability”.  In general, the conditions in our paper are Lipschitz continuous conditions, capturing smoothness of the changes. We chose to use the term “stability” because these conditions both characterize properties regarding system evolution, similar to what “stability” conventionally means in control theory. We will make it clear from the beginning that this does not carry the same meaning as in control theory.
>
> - $Weakness 3$
>
> RE: Thanks for your suggestions on the theorem statement. We will adjust accordingly.
>
> - $Weakness 4$
>
> RE: Thank you for your comment. We will rescale the axes properly.
>
> - $Weakness 5$
>
> RE: Yes, for discrete state-action spaces with countable elements, we can write $\mu_h^{\star}(s, a) = \mathbb{P}_{\xi_1, \pi^{\star}}[S_h = s, A_h = a]$. However, since we are considering RL on continuous spaces, we adopted the notations in the paper to maintain the generality of the definition. We will add a footnote in the revision to clarify this.
>
> - $Question$
>
> RE: That is a great point. It is always desirable to see how theory can develop into efficient new algorithms, and this will definitely be our next step. Regarding the efficacy of FQI, we believe the rate in terms of $n$ or $T$ should be optimal. However, there is still plenty of room to consider other factors. For instance, our theory might help devise new strategies for tackling distributional shift in offline data or improving exploration in online RL. These new methods could significantly reduce the covariate shift or dimensionality issues that also appear in the value sub-optimality upper bound.

---

> > ### Comment · Reviewer_PVTM · 2024-08-12
> >
> > Thank you for your detailed response. I will keep my rating as is.

---

> ### Comment · Area_Chair_QLuK · 2024-08-11
> **Please respond to the authors**
>
> Hello reviewer PVTM: The authors have responded to your comments. I would expect you to respond in kind.

---

### Official Review · Reviewer_gsVp · 2024-07-11

**Soundness:** 4
**Presentation:** 3
**Contribution:** 3
**Rating:** 7
**Confidence:** 3

**Summary:**

This paper demonstrates how sample complexity can be improved under some conditions in offline and online RL.

It considers an episodic MDP framework with episodes of length H.

One of the contributions of the paper is that, under specific stability conditions for the MDP, getting Bellman residual errors < \epsilon results in a policy (more precisely a policy sequence) whose value sub-optimality is bounded by a quantity proportional to \epsilon^2.
As a result, in offline RL, algorithms for which \epsilon is proportional to n^{-1/2} (it is proved for ridge-regression-based Fitted Q-iteration in the paper, under some assumptions) result in a sub-optimality that dwindles proportionally to 1/n, and not n^{-1/2} which is the usual result (n is the number of samples).

For online RL, the paper studies a 2-phase algorithm based for on exploration for T0 episodes (similar to the offline case), and a fine-tuning phase based on online RL performing a sequence of rounds (and the number of episodes in round k is twice the number of episodes in round k-1). In this context and with the same assumpions as before, the paper shows that the regret upper bound grows as log(T), T being the total number of episodes.

These results identify cases in which the rates of convergence are significantly faster than the usual theoretical results, and intuitions related to these "fast rate cases" are extensively discussed in the paper, and in particular their links with optimism and pessimism.

An illustration of fast rate RL achieved in practice is shown on the famous Mountain Car problem.

**Strengths:**

- The paper is well-written, with a high level of technicality.

- The results obtained are interesting because they may help shedding light on properties that make RL "easy", and how algorithms could potentially maximize their effectiveness on such easy cases. This is very important because a lot of continuous RL problems are hard mostly because of their high-dimensional and non-linear nature, but the "absorbing regions" of these problems (to re-employ a term used in the paper) often have good smoothness properties and can be categorized as "easy".

- The developments of the paper are well motivated, and there are many discussions trying to make a link between the results and simpler intuitions.

- Although the detailed analyses are based on assumptions of linear function approximation, the framework proposed in the paper is general, and could be applied to more complex contexts.

**Weaknesses:**

- Discussing thoroughly on the qualitative implications of the results is a strength of the paper, but in some sense it can also be a weakness, since it can be tempting to jump to conclusions that are more global than the restricted context in which results have been proven. For instance, considering all the assumptions on which the results depend, statements like "Thus, the pessimism principle can be useful for problems with smaller sample sizes" must be taken with caution.

**Questions:**

- The global approach has similarities with the much simpler result of logarithmic regret obtained in ["Logarithmic Regret Algorithms for Online Convex Optimization", Hazan et al.]. It raises the following: could some of the results or new results (maybe in a non-episodic framework) be obtained as direct consequences of the result in ["Logarithmic Regret Algorithms for Online Convex Optimization", Hazan et al.]? Typically, x would be a policy, and the update on x would depend on a policy gradient (e.g. with REINFORCE). In this case, would there be any relationship between the assumptions of the paper and the strong convexity required by ["Logarithmic Regret Algorithms for Online Convex Optimization", Hazan et al.]? In a similar direction, analyzing the relationships between the proposed approach and the following papers would be interesting:

Lale, S., Azizzadenesheli, K., Hassibi, B., & Anandkumar, A. (2020). Logarithmic regret bound in partially observable linear dynamical systems. Advances in Neural Information Processing Systems, 33, 20876-20888.

Agarwal, N., Hazan, E., & Singh, K. (2019). Logarithmic regret for online control.
Advances in Neural Information Processing Systems, 32.

- Are the required theoretical assumptions valid in the Mountain Car environment?

- Typos:

line 369 "In the initial T0f episodes" => "In the initial T0 episodes"

line 854 "was previously defined (??)"

**Limitations:**

The limitations of the results are well addressed in the paper.

---

> ### Author Rebuttal · Authors · 2024-08-07
>
> Thank you for your appreciation of our paper. We are grateful for your recognition of our theory as both well-motivated and general. Below, we will address each of the points you’ve raised.
>
> $Weakness$
>
> That is a good point; we will be more cautious about the rigor of the discussions.
>
> $Question$
> - "Could some of the results or new results (maybe in a non-episodic framework) be obtained as direct consequences of the result in ["LogarithmicRegret Algorithms for Online Convex Optimization", Hazan et al.]? ... In a similar direction, analyzing the relationships between the proposed approach and thefollowing papers would be interesting: Lale, S. et. al. (NeurIPS 2020), Agarwal, N. et. al. (NeurIPS 2019)"
>
> RE: Thank you for bringing these pieces of literature to our attention.
>
> [Connection of our results to strong convexity]
>
> You are correct; the fast-rate convergence discussed in our paper has strong connections with fast rates in stochastic optimization and (partially observable) linear dynamical systems, as we partially introduced in Section 1.3. Our theory indicates that the fast rate for continuous RL results from the geometry of the function class, sharing the same spirit as Hazan et al. (2007), Lale et al. (NeurIPS 2020), and Agarwal et al. (NeurIPS 2019), although the “strong convexity” may pertain to different components in different models. We will provide a more thorough discussion on these connections in the revision.
>
> To clarify the connection further, in our paper, we used the linear function approximation example to show that the two stability conditions in the main theorem result from local curvature/strong convexity. In Sections 3.1 and 3.2, where we discuss how curvature implies fast rates, we mentioned in line 253 that “these two (curvature) inequalities arise naturally from a sensitivity analysis of maximizing a linear objective function over a constraint set defined by the feature mapping.” Our understanding is that this sensitivity analysis is closely related to the principles behind Hazan et al. (2007), Lale et al. (NeurIPS 2020), and Agarwal et al. (NeurIPS 2019). Further exploration of these connections is needed.
>
>
> [Generalization to policy-based methods]
>
> Thank you for the great suggestion. It is indeed interesting and stimulating to consider whether the framework can be extended to policy-based or actor-critic methods, such as REINFORCE. By parameterizing the policy class, it is possible that some geometric structures in the parameter space could guarantee super efficiency. For those policy optimization settings, the occupation measure stability condition in our paper—interpreted as the stability of the system under policy perturbation—may still be useful and essential.
>
> - Validation of our theory in the Mountain Car environment.
>
> RE: Thank you for your advice. It will help complete our discussion. We plan to include the following paragraph in the revised version:
>
> In the numerical example of the mountain car problem, we employed linear features defined by trigonometric and polynomial functions. This choice resulted in the feature set $ \Phi(s) = ${$\boldsymbol{\phi}(s,a) \mid a \in \mathcal{A}$} forming a one-dimensional smooth manifold within the $\mathbb{R}^{3000}$ space. The manifold has positive curvature around the point $\boldsymbol{\phi}(s, \pi^{\star}(s))$, thereby validating our arguments (Curv 1) and (Curv 2).
>
> - Typos: Thanks for catching those typos. We will revise them accordingly.

---

> > ### Comment · Reviewer_gsVp · 2024-08-12
> >
> > Thank you for these answers and clarifications. My recommendation of acceptance remains.

---

> ### Comment · Area_Chair_QLuK · 2024-08-11
> **Please respond to the authors**
>
> Hello reviewer gsVp: The authors have responded to your comments. I would expect you to respond in kind.

---

### Decision · Program_Chairs · 2024-09-25

**Decision:**

Accept (poster)

**Comment:**

The paper presents conditions under which continuous (in states and actions) reinforcement learning converges rapidly in both online and offline settings. The reviewers are uniformly in agreement that the paper is deserving of acceptance. I have read the paper myself and agree that the results are interesting, and to my knowledge novel. I am confident that the results will be of interest to the NeurIPS community.